# Variance-Reduced Long-Term Rehearsal Learning with Quadratic Programming Reformulation

**Wen-Bo Du, Tian Qin, Tian-Zuo Wang, Zhi-Hua Zhou**

National Key Laboratory for Novel Software Technology, Nanjing University, China
School of Artificial Intelligence, Nanjing University, China
{duwb, qint, wangtz, zhouzh}@lamda.nju.edu.cn

## Abstract

In machine learning, a critical class of decision-making problems involves *Avoiding Undesired Future* (AUF): given a predicted undesired outcome, how can one make decision about actions to prevent it? Recently, the *rehearsal learning* framework has been proposed to address AUF problem. While existing methods offer reliable decisions for single-round success, this paper considers long-term settings that involve coordinating multiple future outcomes, which is often required in real-world tasks. Specifically, we generalize the AUF objective to characterize a long-term decision target that incorporates cross-temporal relations among variables. As directly optimizing the *AUF probability* $\mathbb{P}_{\mathrm{AUF}}$ over this objective remains challenging, we derive an explicit expression for the objective and further propose a quadratic programming (QP) reformulation that transforms the intractable probabilistic AUF optimization into a tractable one. Under mild assumptions, we show that solutions to the QP reformulation are equivalent to those of the original AUF optimization, based on which we develop two novel rehearsal learning methods for long-term decision-making: (i) a *greedy* method that maximizes the single-round $\mathbb{P}_{\mathrm{AUF}}$ at each step, and (ii) a *far-sighted* method that accounts for future consequences in each decision, yielding a higher overall $\mathbb{P}_{\mathrm{AUF}}$ through an $L/(L+1)$ variance reduction in the AUF objective. We further establish an $\mathcal{O}(1/\sqrt{N})$ excess risk bound for decisions based on estimated parameters, ensuring reliable practical applicability with finite data. Experiments validate the effectiveness of our approach.

## 1 Introduction

Machine learning (ML) methods have demonstrated remarkable success in diverse real-world prediction tasks [1]. Instead of solely focusing on the prediction, Zhou [2] emphasizes another important issue, *i.e.*, if an ML model predicts that something undesired is going to occur, how to find effective actions to prevent it from happening. This problem is known as *avoiding undesired future (AUF)* [2].

Consider a portfolio management system as an example, which employs an ML model using economic indicators ($\mathbf{X}$) to predict monthly portfolio returns ($\mathbf{Y}$). If a predicted return is undesirably low, an ideal system would have the ability of suggesting to alter allocation weights ($\mathbf{Z}$) across asset classes (e.g., stocks, bonds) to enhance profitability. Due to expensive transaction costs, alterations must be carefully justified. Let $\mathcal{S}$ denote the desired region specified by decision-makers, AUF can be framed as finding alterations $\mathbf{z}^{\xi}$ that maximize the probability of $\mathbf{Y} \in \mathcal{S}$. In real-world tasks, decisions are often temporally coupled, *e.g.*, aggressive risk-seeking might boost short-term returns while increasing future vulnerability. This motivates a more practical AUF goal: finding a sequence of *alterations* $(\mathbf{z}_i^{\xi}, \ldots, \mathbf{z}_{i+L}^{\xi})$ to maximize the probability that the aggregated outcome $\bar{\mathbf{Y}} \triangleq \frac{1}{L+1} \sum_i \mathbf{Y}_i$ falls within $\mathcal{S}$. This long-term formulation captures cross-period trade-offs, promotes sustainable performance, and encompasses the single-round setting as a special case ($L = 0$).

39th Conference on Neural Information Processing Systems (NeurIPS 2025).

Generally, the AUF probability $\mathbb{P}_{\mathrm{AUF}}(\cdot)$ measures the likelihood that $\cdot$ falls into the desired region $\mathcal{S}$ conditioned on the selected alterations. However, the dependence of $\mathbb{P}_{\mathrm{AUF}}$ on alterations is typically inaccessible and highly complex, particularly in long-term settings, which makes the identification of effective decisions challenging. In such cases, leveraging the structural relations among variables $\{\mathbf{X}, \mathbf{Z}, \mathbf{Y}\}$ be-

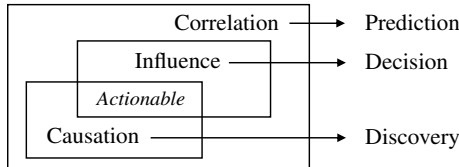

Figure 1: Relations between correlation, causation, and influence (reproduced from Zhou [2]).

comes essential. While *correlation* is often sufficient for prediction, it is generally inadequate for guiding alterations [2]. *Causal relations*, in contrast, would be more informative, but identifying them usually requires restrictive and untestable assumptions and may involve factors that are not actionable [3, 4]. Recognizing that *correlation* is insufficient and *causation* is somewhat too luxurious to decision-making, Zhou [5] propose the *influence relation* as the foundational concept for decision-making, which is different from correlation and causation, as depicted in Fig. 1; and propose to study *rehearsal learning* approaches to identify and exploit influence relation.

Building on this, *rehearsal learning* approaches [6, 7] have been developed and been shown effective in single-round AUF, where the objective is to ensure $\mathbf{Y} \in \mathcal{S}$ for immediate outcomes. These methods typically leverage distributional properties of $\mathbf{Y}$ to reformulate the problem as identifying alterations that satisfy a constraint $\mathbb{P}_{\mathrm{AUF}} \geq \tau$, thereby circumventing the intractable probabilistic optimization. Extending this idea into long-term settings, however, introduces several fundamental challenges: (i) Temporal dependencies among outcomes $\mathbf{Y}_i$ across different rounds complicate the analysis of $\mathbb{P}_{\mathrm{AUF}}$, as one must now account for the aggregation of correlated $\mathbf{Y}_i$ rather than a single outcome; (ii) Long-term decision-making demands alterations that closely approximate the optimal $\mathbb{P}_{\mathrm{AUF}}$ at each round, since even small deviations can accumulate into substantial performance degradation over time; (iii) The dimensionality of alteration sequence grows with the horizon length, substantially increasing computational complexity. These challenges lie beyond the capabilities of existing approaches.

To enable the rehearsal learning framework in long-term AUF scenarios, new methods are demanded to address the aforementioned challenges. In this paper, we first generalize the decision objective to a long-term aggregated form, which subsumes existing formulations as special cases. Under this generalized formulation, we show that the aggregated objective can be explicitly decomposed into: (i) the cumulative effect of selected alterations and (ii) the aggregated noise unrelated to alterations, making it feasible to design decision-making strategies that bypass the explicit modeling of temporal dependencies among outcomes $\mathbf{Y}_i$. Leveraging this decomposition, we propose a quadratic programming (QP) reformulation of the original intractable optimization. This reformulation yields a tractable optimization target, and scalable to long time horizons. We further establish theoretical guarantees for the optimality of the QP solution under mild assumptions, which effectively mitigates the error accumulation issues inherent in the aforementioned constraint-based formulations. Notably, we make an independent contribution that our result applies to the class of log-concave noise distributions, which generalizes beyond the Gaussian noise assumption [6–8]. Based on the results above, we develop two new rehearsal-based methods focused on long-term outcomes: (i) a greedy approach that maximizes the single-round $\mathbb{P}_{\mathrm{AUF}}$ at each step, outperforming existing rehearsal-based methods, and (ii) a far-sighted approach that anticipates the future consequences of current decisions, consistently achieving a higher overall $\mathbb{P}_{\mathrm{AUF}}$ by attaining an $L/(L+1)$ variance reduction rate in the AUF objective. Finally, recognizing that true structural parameters are often unknown in practice, we establish an excess risk bound for decisions made using parameters estimated from observational data. Experiments validate our theory and demonstrate the effectiveness and efficiency of our approach.

Our contributions can be summarized as follows:

1. We generalize the AUF problem to a long-term perspective and derive an explicit expression for the objective, offering greater practical flexibility while subsuming existing formulations.

2. We introduce a QP reformulation, making the AUF optimization tractable and efficiently solvable. The optimality of the QP solution is then established under mild assumptions.

3. We develop two new rehearsal-based algorithms based on the QP reformulation, and the far-sighted one can achieve better performance in long-term scenarios due to its variance-reduction property.

4. We establish an $\mathcal{O}(1/\sqrt{N})$ bound on the excess risk incurred when using estimated parameters instead of the true ones, guaranteeing reliable performance in practical scenarios.

## 2 Preliminaries

A probabilistic graphical model, termed the structural rehearsal model (SRM), was proposed by Qin et al. [6] to characterize influence relations among variables in the AUF problem. The SRM comprises a set of rehearsal graphs and corresponding structural equations, denoted as $\{\langle G_t, \boldsymbol{\theta}_t \rangle\}$, where $t$ represents the decision round. Each graph $G_t = (\mathbf{V}_t, \mathbf{E}_t)$ encodes the variables $\mathbf{V}_t$ and induced edges $\mathbf{E}_t$ at round $t$, while the parameter set $\boldsymbol{\theta}_t$ includes both the generating parameters associated with $\mathbf{E}_t$ and the noise parameters related to $\mathbf{V}_t$. The formal definition of SRM is provided in Appx. B. Notably, existing research on rehearsal learning considers only the graph structure and influence relations within a single decision round, neglecting cross-round influences. That is, they assume no edges exist between $V_{t_1}^i$ and $V_{t_2}^j$ for $t_1 \neq t_2$.

In this work, we extend the SRM to a multivariate time series setting $\{\mathbf{V}_t\}_{t \in [0,K]}$ that captures both lagged and contemporaneous influences within a finite time window $K$. Specifically, let $\mathbf{E}_t^{\text{cross}}$ denote all cross-time edges pointing into variables at time $t$, *i.e.*, edges from $V_{t'}^i$ to $V_t^j$ for all $0 \leq t' < t$ and $1 \leq i, j \leq |\mathbf{V}|$. The complete graph is then defined as $\mathbf{G}_K = (\cup_{t \in [0,K]} \mathbf{V}_t, \cup_{t \in [0,K]} (\mathbf{E}_t \cup \mathbf{E}_t^{\text{cross}}))$, as illustrated in Fig. 2. $\mathbf{G}_K$ models the qualitative influence relations among variables across different time steps, where $\mathbf{V}_t$ represents the variable set in the AUF problem and $\mathbf{E}_t \cup \mathbf{E}_t^{\text{cross}}$ captures the influence relations among variables at time $t$. There are two types of edges: (i) a directed edge $V_{t'}^i \rightarrow V_t^j$ ($i \neq j, t' \leq t$) indicates that $V_{t'}^i$ unilaterally influences $V_t^j$; whereas (ii) a bidirectional edge $V_t^i \leftrightarrow V_t^j$ ($i \neq j$) signifies mutual influence between $V_t^i$ and $V_t^j$. For instance, birth rates unilaterally influence demographic structures, whereas supply and demand in an idealized market exhibit mutual dependence, as changes in one directly affect the other.

To quantitatively describe these influences, let $\boldsymbol{\Theta}_t = \boldsymbol{\theta}_t \cup \boldsymbol{\theta}_t^{\text{cross}}$ denote the extended parameter set of $\mathbf{G}_K$ in time $t$, where $\boldsymbol{\theta}_t^{\text{cross}}$ corresponds to the parameters associated with $\mathbf{E}_t^{\text{cross}}$. The structural equations governing the variables $V_t^j$ can then be parameterized by $\boldsymbol{\Theta}_t$:

$$V_t^j := f_t^j \left( \text{PA}_t^j, \varepsilon_t^j; \boldsymbol{\Theta}_t^j \right) \quad \text{for} \quad 0 \leq t \leq K \text{ and } 1 \leq j \leq |\mathbf{V}|, \tag{1}$$

where $\text{PA}_t^j \triangleq \{u \mid u \rightarrow V_t^j \text{ in } \mathbf{G}_K\}$ represents parents of $V_t^j$, and $\varepsilon_t^j$ denote the noise. The structural function $f_t^j(\cdot)$ and the probability density function (PDF) of $\varepsilon_t^j$ are parameterized by $\boldsymbol{\Theta}_t^j$. In this work, we make the first-order Markov assumption of the time series for simplified expression (can be straightforwardly extend to general cases), under which we focus on a basic yet essential class of the AUF problem, characterized by stationary linear structural equations $f_t^j$s in Eq. (1), *i.e.*,

$$\mathbf{V}_t = \mathbf{A}\mathbf{V}_t + \mathbf{B}\mathbf{V}_{t-1} + \boldsymbol{\varepsilon}_t, \quad \text{for} \quad 1 \leq t \leq K, \tag{2}$$

where $\mathbf{A}$ and $\mathbf{B}$ are determined by $\boldsymbol{\Theta}_t$ and satisfy $\rho((\mathbf{I} - \mathbf{A})^{-1}\mathbf{B}) < 1$ ($\rho(\cdot)$ denotes the maximal absolute eigenvalue) to ensure stationarity [9]; and where $\boldsymbol{\varepsilon}_t$ follows a white noise process with $\mathbb{E}[\boldsymbol{\varepsilon}_t] \triangleq \mathbf{0}$, and $\mathbb{E}[\boldsymbol{\varepsilon}_t \boldsymbol{\varepsilon}_s^\mathsf{T}] \triangleq \mathbf{0}$ for any $t \neq s$, otherwise an invariant $\boldsymbol{\Sigma}$. In this work, the noise is assumed to follow a symmetric log-concave distribution, *i.e.*, the PDF $f_{\boldsymbol{\varepsilon}}(\mathbf{v})$ is log-concave and satisfies $f_{\boldsymbol{\varepsilon}}(\mathbf{v}) = f_{\boldsymbol{\varepsilon}}(-\mathbf{v})$ for all $\mathbf{v}$. This family includes many common zero-mean distributions, such as Gaussian, uniform, and Laplace. Finally, the decision-making task focuses on identifying suitable *alterations* $Rh(\mathbf{z}^\xi)$, which disrupts existing influence links, as detailed in Appx. B.

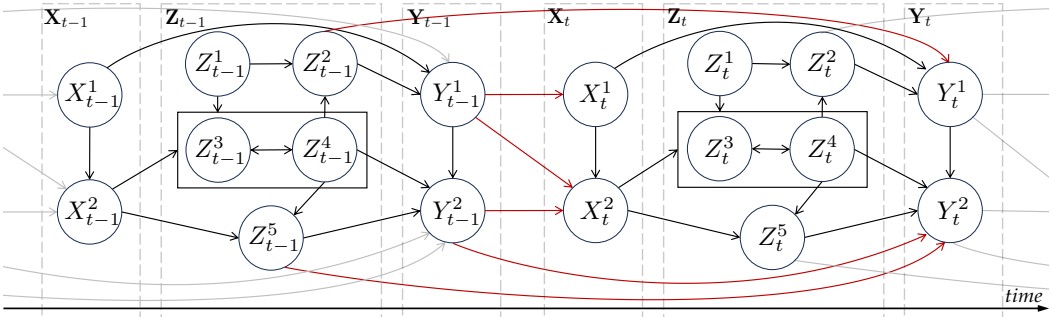

Figure 2: An example illustrating variables and edges at rounds $t - 1$ and $t$. Red edges indicate cross-round influence relations. The variable sets $\mathbf{X}, \mathbf{Z}, \mathbf{Y}$ are enclosed by dashed lines.

# 3 The proposed approach

In this section, we first present a generalized formulation of the AUF problem in Sec. 3.1, incorporating long-term aggregation over target variables. In Sec. 3.2, we derive an explicit expression for the aggregated AUF objective w.r.t. the alteration sequence and propose a QP reformulation with theoretical guarantees of optimality, offering a new perspective for rehearsal learning. Building on this theoretical foundation, in Sec. 3.3, we propose two novel rehearsal-based algorithms for solving the formulated problem. Finally, in Sec. 3.4, we establishe theoretical guarantees on both the variance reduction and the excess risk. All proofs of the theoretical results are provided in Appx. C.

## 3.1 Formulation

We treat the AUF problem as a sequence of immediate decision-making processes during a certain time period $t, \cdots, t + T$. In each decision round, the decision maker should make timely decisions based on all available information; while the final decision target is to maximize the probability of $\bar{\mathbf{Y}} \in \mathcal{S}$ within this time period. Each decision round involves two critical time points: the moment when the ML prediction is made after observing $\mathbf{X}$ and the moment just before the generation of the concerned outcome $\mathbf{Y}$, as illustrated in Fig. 2. Specifically, in a given decision round $t$, the decision maker first observes $\mathbf{X}_t = \mathbf{x}_t$ and incorporates it into the historical dataset, defined as $D_t \triangleq \{\mathbf{x}_t\} \cup \{\mathbf{x}_i, \mathbf{z}_i^\xi, \mathbf{y}_i\}_{i<t}$. The ML model then predicts the values of the target variables $\mathbf{Y}_t, \ldots, \mathbf{Y}_{t+T}$ over a future horizon. Let $\mathcal{S} \subseteq \mathbb{R}^{|\mathbf{Y}|}$ denote the predefined desired region for the target variables $\bar{\mathbf{Y}}$, if the predicted values fail to satisfy the predefined criterion, *i.e.*, $\bar{\mathbf{Y}} \notin \mathcal{S}$, corrective decisions should be made to prevent undesirable outcomes. This decision-making problem in time $t$ can be formally expressed as the following probabilistic optimization problem:

$$\underset{\{\mathbf{z}_i^\xi\}_{i=t}^{t+T}}{\arg\max} \quad \mathbb{P}\left(\frac{1}{T+1}\sum_{i=t}^{t+T}\mathbf{Y}_i \in \mathcal{S} \;\Big|\; D_t, Rh(\mathbf{z}_t^\xi, \cdots, \mathbf{z}_{t+T}^\xi)\right).$$

$$\text{s.t.} \quad \mathbf{z}_i^\xi \in \Delta(\mathbf{Z}) \quad \text{for} \quad i = t, \cdots, t+T, \tag{3}$$

where $T + 1$ is the length of the remaining target time horizon specified by the decision-maker, $\Delta(\mathbf{Z})$ represents the feasible domain of each alteration $\mathbf{z}_i^\xi$, and the rehearsal operator $Rh(\cdot)$ denotes the execution of the alteration sequence $\mathbf{Z}_i \overset{a}{=} \mathbf{z}_i^\xi$ for $i = t, \ldots, t+T$. This operator modifies the local graph structure by disrupting specific influence links, as detailed in Appx. B.

Previous studies primarily focused on a simplified setting where $T \triangleq 0$, and assumed a Gaussian distribution on the target probability [6, 10]. In this paper, we consider a more general formulation where the AUF objective is defined over aggregated target variables across a time period, which can reduce to the prior setting. This generalization is commonly encountered in practical scenarios. For instance, when $\mathbf{Y}_i$ denotes the annual GDP growth rate, the objective might be to maintain a high average rate over several consecutive years. Additionally, we generalize the noise distribution to a symmetric log-concave family, which includes Gaussian, uniform, and Laplace distributions as special cases, thereby expanding the practical applicability.

The main challenges of the formulated problem in Eq. (3) are twofold: (i) The target variables $\mathbf{Y}_i$ across time are generally dependent, making it substantially more difficult to characterize the distribution and probabilistic properties of their aggregation compared to the single-time case, particularly under non-Gaussian noise. (ii) The dimensionality of decision variables grows with the horizon length $T$, introducing additional computational overhead and rendering direct extensions of prior rehearsal learning methods infeasible. For instance, the asymptotic solution by sampling methods in Qin et al. [6] becomes intractable in polynomial time unless the decision variable is singleton (i.e., $|\mathbf{z}^\xi| = 1$), complicating its application to the aggregated setting.

## 3.2 QP reformulation of the AUF problem

In this subsection, we first derive a functional expression for the aggregated target variables w.r.t. the selected alteration sequence and the noise sequence. Based on this expression, we propose a QP reformulation of AUF problem in Eq. (3), which is computationally efficient to solve. Furthermore, we establish a theoretical guarantee for the optimality of the QP solution under mild assumptions.

To maximize $\mathbb{P}_{\text{AUF}}$, the first step is to derive the functional expression for the AUF objective $\bar{\mathbf{Y}}$ w.r.t. the selected alteration sequence. Recall from Sec. 3.1 that decisions are made after observing $\mathbf{X}_t = \mathbf{x}_t$ and just before $\mathbf{Z}_t$ occurs. In this time point, the aggregation of $\mathbf{Y}$ can be derived as follows:

**Proposition 3.1.** *Let* $\tilde{\mathbf{z}}_t^\xi$ *and* $\tilde{\mathbf{e}}_t$ *denote* $\left[\mathbf{z}_{t+T}^{\xi}{}^\mathsf{T}, \mathbf{z}_{t+T-1}^{\xi}{}^\mathsf{T}, \cdots, \mathbf{z}_t^{\xi\mathsf{T}}\right]^\mathsf{T}$ *and* $\left[\boldsymbol{\varepsilon}_{t+T}^\mathsf{T}, \boldsymbol{\varepsilon}_{t+T-1}^\mathsf{T}, \cdots, \boldsymbol{\varepsilon}_t^\mathsf{T}\right]^\mathsf{T}$. *Given* $\boldsymbol{\Theta}$ *(*$\supseteq \mathbf{A}, \mathbf{B}$ *in Eq.* (2)*),* $D_t$ *(including* $\mathbf{v}_{t-1}$ *and* $\mathbf{x}_t$*) and alteration sequence* $\tilde{\mathbf{z}}_t^\xi$*, it holds that:*

$$\frac{1}{T+1}\sum_{i=t}^{t+T}\mathbf{Y}_i = \mathbf{M}\mathbf{x}_t + \mathbf{N}\mathbf{v}_{t-1} + \mathbf{H}\tilde{\mathbf{z}}_t^\xi + \mathbf{F}\tilde{\mathbf{e}}_t, \tag{4}$$

*where* $\mathbf{M} \in \mathbb{R}^{|\mathbf{Y}|\times|\mathbf{X}|}, \mathbf{N} \in \mathbb{R}^{|\mathbf{Y}|\times|\mathbf{V}|}, \mathbf{H} \in \mathbb{R}^{|\mathbf{Y}|\times(T+1)|\mathbf{Z}|}$ *and* $\mathbf{F} \in \mathbb{R}^{|\mathbf{Y}|\times(T+1)|\mathbf{V}|}$ *are constant matrices based on parameters* $\boldsymbol{\Theta}$ *and time period* $T$*, while* $\boldsymbol{\varepsilon}_t, \cdots, \boldsymbol{\varepsilon}_{t+T}$ *are i.i.d. noise vectors.*

Although the target variables $\mathbf{Y}_i$ are dependent across different $i \in [0, T]$, Prop. 3.1 shows that their aggregation can be explicitly decomposed into: (i) a mean component, determined by the value of alteration variables $\tilde{\mathbf{z}}_t^\xi$ given the observed $\mathbf{x}_t$ and $\mathbf{v}_{t-1}$; and (ii) a stochastic component, governed by the aggregated temporal noise, with dependence captured by $\mathbf{F}\tilde{\mathbf{e}}_t$. Moreover, Prop. 3.2 shows that the distribution family can be preserved under symmetric log-concave noise.

**Proposition 3.2.** *Let* $\mathbf{e} \triangleq \mathbf{F}\tilde{\mathbf{e}}_t$ *denote the aggregated noise term as defined in Eq.* (4). *If* $\varepsilon_i$ *follows a symmetric log-concavely distribution, it always holds that* $\mathbf{e}$ *is also log-concavely distributed and symmetric about the origin, for any finite time point* $t \in \mathbb{Z}_+$ *and finite time window* $T \in \mathbb{Z}_+$.

As established in Prop. 3.2, the distribution family is preserved under aggregation of the target variables $\mathbf{Y}$, thereby retaining desirable properties of log-concave distributions, such as the unimodality [11]. Nevertheless, directly solving the probabilistic optimization in Eq. (3) remains challenging, as the resulting PDF of the aggregated outcome can still exhibit considerable complexity.

To avoid directly solving the probabilistic optimization, existing rehearsal-based methods typically reformulate it by imposing a constraint of the form $\mathbb{P}_{\text{AUF}} \geq \tau$, and then search for plausible alterations that satisfy this constraint. Specifically, Qin et al. [6] propose a Monte Carlo (MC) method that samples from the distribution $P(\mathbf{Y} \mid \mathbf{x}, Rh(\mathbf{z}^\xi))$, and approximates the solution by requiring that a proportion greater than $\tau$ of the samples satisfy $\mathbf{Y}_i \in \mathcal{S}$. In contrast, Du et al. [7] search for a probabilistic region $\mathcal{P}$ such that $\mathbb{P}(\mathbf{Y} \in \mathcal{P} \mid \mathbf{x}, Rh(\mathbf{z}^\xi)) = \tau$, and ensure that all points within $\mathcal{P}$ fall into the desired region $\mathcal{S}$ after performing the alteration. These methods are theoretically grounded under Gaussian noise assumption to ensure $\mathbb{P}_{\text{AUF}} \geq \tau$, but the optimality can not be guaranteed.

Instead, we leverage the unimodality property of log-concave distributions to guide the decision-making process. Recall that the goal is to shift as much probability mass as possible into the desired region $\mathcal{S}$. As shown in Eq. (4), the alterations affect only the mean of the target variables, while the variance is determined by $\mathbf{F}\tilde{\mathbf{e}}_t$. Therefore, if $\mathcal{S}$ is a closed region in $\mathbb{R}^{|\mathbf{Y}|}$, then adjusting $\mathbf{z}^\xi$ to move the peak of the unimodal distribution toward the center of mass of $\mathcal{S}$ can effectively increase the likelihood of favorable outcomes. Let $\mathbf{M}, \mathbf{N}, \mathbf{H}$ be defined as in Prop. 3.1, and let $\mathbf{s}$ denote the center of mass of $\mathcal{S}$. This heuristic motivates a deterministic QP reformulation that minimizes the distance between the shifted mean and the region $\mathcal{S}$, formulated as:

$$\underset{\tilde{\mathbf{z}}_t^\xi}{\arg\min} \quad \left\|\mathbf{M}\mathbf{x}_t + \mathbf{N}\mathbf{v}_{t-1} + \mathbf{H}\tilde{\mathbf{z}}_t^\xi - \mathbf{s}\right\|_2^2. \tag{5}$$

While this heuristic idea is simple to implement, there remains a risk that the resulting solution may deviate significantly from the true maximizer of the $\mathbb{P}_{\text{AUF}}$, even in simple scenarios (see Appx. A for a discussion). To clarify the conditions under which the QP reformulation is effective, we introduce the following assumptions and establish the optimality of the reformulation under these conditions.

**Assumption 3.3.** The following assumptions constrain the system and the problem formulation.

1. (**Linear system**) The structural functions $f_i$ in Eq. (1) are linear with additive noise terms $\varepsilon_t^j$, where $\varepsilon_t$ follows a symmetric log-concave distribution, as defined in Eq. (2).

2. (**Unique targets**) Let $\boldsymbol{\mu}_t = \mathbb{E}[1/(T+1)\sum_{i=t}^{t+T}\mathbf{Y}_i \mid D_t, Rh(\tilde{\mathbf{z}}_t^\xi)]$ denote the mean vector of the AUF target as defined in Eq. (4), which is an affine function of $\tilde{\mathbf{z}}_t^\xi$. Given $t$ and $T$, vector $\partial\mu_t^i/\partial\tilde{\mathbf{z}}_t^\xi$ cannot be expressed as a linear combination of the set $\{\partial\mu_t^j/\partial\tilde{\mathbf{z}}_t^\xi\}_{j\neq i}$ for any $1 \leq i \leq |\mathbf{Y}|$.

3. (**Symmetric $\mathcal{S}$**) The desired region $\mathcal{S}$ in Eq. (3) is a centrally symmetric (about $\mathbf{s}$) convex region.

**Remark.** Note that the first two assumptions pertain to the underlying system, while the last one concerns the AUF problem itself. Specifically, the **linear system** assumption permits a broader class of symmetric log-concave noise distributions, generalizing prior works that often assume Gaussian noise for theoretical tractability. The **unique targets** assumption is also reasonable: if it were violated, *i.e.*, $\exists i$ such that $\partial \mu_t^i / \partial \tilde{\mathbf{z}}_t^\xi$ can be expressed as a linear combination of $\{\partial \mu_t^j / \partial \tilde{\mathbf{z}}_t^\xi\}_{j \neq i}$, then the variable $Y^i$ would be redundant in the decision-making process, as its constraints would be fully captured by those on the other target variables, and $Y^i$ could therefore be omitted. Finally, since the desired region $\mathcal{S}$ is defined by the decision-maker, it is typically specified manually. While previous studies primarily consider convex polytopes as the region shape, our **symmetric $\mathcal{S}$** assumption accommodates additional shapes such as circular and elliptical regions, though non-symmetric ones are excluded. This assumption is practical in many settings. For instance, interval constraints on each dimension of $\bar{\mathbf{Y}}$ (i.e., $a_i \leq \bar{Y}^i \leq b_i$ for each $i$) naturally satisfy it and are common in real-world applications. Building on these assumptions, we establish theoretical guarantees for the QP.

**Theorem 3.4.** *Let $\Delta(\mathbf{Z}) = \mathbb{R}^{|\mathbf{z}^\xi|}$ and $\tilde{\mathbf{z}}_t^\star$ denote the solution to the QP defined in Eq.* (5)*. Under Ass.* 3.3 *with any finite $t, T \in \mathbb{Z}_+$, the following inequality holds for any alternative $\tilde{\mathbf{z}}_t^a$:*

$$\mathbb{P}\left(\frac{1}{T+1}\sum_{i=t}^{t+T}\mathbf{Y}_i \in \mathcal{S} \;\Big|\; D_t, Rh(\tilde{\mathbf{z}}_t^\star)\right) \geq \mathbb{P}\left(\frac{1}{T+1}\sum_{i=t}^{t+T}\mathbf{Y}_i \in \mathcal{S} \;\Big|\; D_t, Rh(\tilde{\mathbf{z}}_t^a)\right).$$

Thm. 3.4 provides an optimality guarantee for decisions made (by Eq. (5)) immediately after observing $\mathbf{X}_t = \mathbf{x}_t$ in round $t$. This solution is not necessarily globally optimal from the future perspective at time $t + T$, since less information is available at time $t$ compared to $t + T$. Nevertheless, Thm. 3.4 exhibits more favorable applicability in prior AUF settings that consider only the target outcome $\mathbf{Y}$ of the current decision round (*i.e.*, $T = 0$), because the solution in this case coincides with the global optimum and thus outperforms existing methods. Moreover, even when $T \neq 0$, the solution can also be viewed as the best possible decision at time $t$ given all available information. This naturally motivates an iterative method that solves for $\tilde{\mathbf{z}}_t^\star$ at each round and executes only the current $\mathbf{z}_t^\xi$. In what follows, we formally propose such two new rehearsal-learning methods based on Thm. 3.4.

### 3.3 Rehearsal learning methods based on the QP reformulation

In this subsection, we utilize aforementioned results to develop two new rehearsal learning algorithms for solving the AUF problem defined in Eq. (3), including a greedy algorithm and a far-sighted one.

**A greedy approach.** Recall that the goal of AUF is to shift the probability mass of the average target variables $\bar{\mathbf{Y}}$ over a time period into the desired region $\mathcal{S}$ as much as possible. A straightforward approach is to greedily maximize the probability of $\mathbf{Y}_i \in \mathcal{S}$ at each round $i$, which intuitively leads to a relatively high probability of $\bar{\mathbf{Y}} \in \mathcal{S}$ as well. This procedure is summarized in Alg. 1. Since we focus on single-round optimality in each decision round, we set $T = 0$ and compute the invariant ma-

---

**Algorithm 1** GMuR (Greedy Multi-round Rehearsal)
**Input:** start/end time $t_0/t_e$, SRM para. $\boldsymbol{\Theta}$, desired region $\mathcal{S}$
1: Determine the symmetric center $\mathbf{s}$ of desired region $\mathcal{S}$
2: Compute $\mathbf{M}, \mathbf{N}, \mathbf{H}$ in Eq. (4) by $T = 0$ and $\boldsymbol{\Theta}$
3:               ▷ Computation formula in Eq. (9), Appx. C
4: Let $\mathbf{H}_{\text{ha}}$ denote the matrix $\mathbf{H}^\mathsf{T}(\mathbf{H}\mathbf{H}^\mathsf{T})^{-1}$
5: Compute $\mathbf{H}_1 = \mathbf{H}_{\text{ha}}\mathbf{M}, \mathbf{H}_2 = \mathbf{H}_{\text{ha}}\mathbf{N}, \mathbf{s}_H = \mathbf{H}_{\text{ha}}\mathbf{s}$
6: **for** $t = t_0$ **to** $t_e$ **do**
7:     Observing $\mathbf{v}_{t-1}$ and $\mathbf{x}_t$
8:     Execute $\mathbf{z}_t^\xi = \mathbf{s}_H - \mathbf{H}_1\mathbf{x}_t - \mathbf{H}_2\mathbf{v}_{t-1}$
9:     Receive $\mathbf{y}_t$ and set $\mathcal{D}_{t+1} = \mathcal{D}_t \cup \{\mathbf{x}_t, \mathbf{z}_t^\xi, \mathbf{y}_t\}$
**Output:** suggested alterations $\{\mathbf{z}_t^\xi\}_{t=t_0}^{t_e}$

---

trices $\mathbf{M}, \mathbf{N}, \mathbf{H}$ using the formula defined in Eq. (9), Appx. C. Then, in each decision round from time $t_0$ to $t_e$, the alteration $\mathbf{z}_t^\xi$ is chosen by selecting a solution of the QP reformulation defined in Eq. (5). Note that in this case, the greedy nature of the approach implies $\mathbf{z}_t^\xi \triangleq \tilde{\mathbf{z}}_t^\star$ because $T = 0$.

By using the GMuR approach described in Alg. 1, Thm. 3.4 guarantees that each alteration $\mathbf{z}_t^\xi$ maximizes the probability of $\mathbf{Y}_t \in \mathcal{S}$. This further ensures that the GMuR approach outperforms prior rehearsal-based methods under Ass. 3.3, since those methods are specifically designed for the single-round AUF setting, which also aims to optimize the probability of $\mathbf{Y}_t \in \mathcal{S}$. Moreover, this method is computationally efficient as the matrices $\mathbf{M}, \mathbf{N}, \mathbf{H}$ remain invariant across rounds due to the fixed horizon $T = 0$, resulting in an overall time complexity of $\mathcal{O}(|\mathbf{z}||\mathbf{y}|^2 + (t_e - t_0)|\mathbf{v}||\mathbf{z}|)$, which outperforms existing rehearsal-based methods [6, 7] in time complexity as discussed in Appx. 4.

**A far-sighted approach.** Although the GMuR approach maximizes the probability that the current target $\mathbf{Y}$ fall into $\mathcal{S}$, it overlooks the sequential nature of the AUF objective, which concerns the aggregate $\bar{\mathbf{Y}}$ over a horizon. To mitigate potential information loss, we propose a far-sighted strategy in Alg. 2, which selects alterations by considering not only current targets but also future ones. Specifically, the method iteratively updates and reweights the matrices $\mathbf{M}, \mathbf{N}, \mathbf{H}$ according to the remaining horizon $T$ and total horizon $L$, repeatedly solves

---

**Algorithm 2** FarMuR (Far-sighted Multi-round Rehearsal)

**Input:** start/end time $t_0/t_e$, SRM para. $\boldsymbol{\Theta}$, desired region $\mathcal{S}$
1: Determine symmetric center $\mathbf{s}$ of $\mathcal{S}$, Set $L = t_e - t_0$
2: **for** $t = t_0$ **to** $t_e$ **do**
3:      Observing $\mathbf{v}_{t-1}$ and $\mathbf{x}_t$, update $T = t_e - t$
4:      Compute $\mathbf{M}, \mathbf{N}, \mathbf{H}$ in Eq. (4) by $T$ and $\boldsymbol{\Theta}$
5:          ▷ Computation formula in Eq. (9), Appx. C
6:      Reweight $(\mathbf{M}, \mathbf{N}, \mathbf{H}) = \frac{T+1}{L+1} \cdot (\mathbf{M}, \mathbf{N}, \mathbf{H})$
7:      Compute $\tilde{\mathbf{z}}_t = \mathbf{H}^{\mathsf{T}}(\mathbf{H}\mathbf{H}^{\mathsf{T}})^{-1}(\mathbf{s} - \mathbf{M}\mathbf{x}_t - \mathbf{N}\mathbf{v}_{t-1})$
8:      Execute $\mathbf{z}_t^{\xi} \triangleq \tilde{\mathbf{z}}_t[: |\mathbf{z}|]$ and drop others
9:      Receive $\mathbf{y}_t$ and set $\mathcal{D}_{t+1} = \mathcal{D}_t \cup \{\mathbf{x}_t, \mathbf{z}_t^{\xi}, \mathbf{y}_t\}$
10:     Update $\mathbf{s} = \mathbf{s} - \mathbf{y}_t/(L+1)$
**Output:** suggested alterations $\{\mathbf{z}_t^{\xi}\}_{t=t_0}^{t_e}$

---

for $\tilde{\mathbf{z}}_t^{\star}$, and executes only the current $\mathbf{z}_t^{\xi}$. After each decision round, it also adjusts the region center $\mathbf{s}$.

The FarMuR approach accounts for the aggregation of $\mathbf{Y}$ over the remaining time horizon at each decision round, resulting in time-varying matrices $\mathbf{M}, \mathbf{N}, \mathbf{H}$. As a consequence, the time complexity of Alg. 2 increases to $\mathcal{O}((t_e - t_0)|\mathbf{z}||\mathbf{y}||\mathbf{v}|)$, higher than that of Alg. 1. However, this increased computational cost leads to improved performance, as discussed in the following subsection.

### 3.4 Theoretical analysis

Recall that Alg. 1 and Alg. 2 shift $\mathbb{E}[\bar{\mathbf{Y}}]$ toward $\mathbf{s}$. As $\bar{\mathbf{Y}}$ follows a unimodal distribution as shown in Prop. 3.2, the probability mass falling within $\mathcal{S}$ is affected not only by $\mathbb{E}[\bar{\mathbf{Y}}]$ but also by $\mathrm{Var}[\bar{\mathbf{Y}}]$.

**Theorem 3.5.** *When $\mathbf{Y}$ is singleton (i.e., $|\mathbf{Y}| = 1$), let $\mathrm{Var}_0 = \mathrm{Var}[Y_t \mid D_t, Rh(\mathbf{z}_t)]$, and let $\mathcal{A}_{t:t+L}$ denote the rehearsal learning process as in Alg. 1 or Alg. 2 that iteratively selects and performs alterations from round $t$ to round $t + L$. Define $\mathrm{Var}_1$ and $\mathrm{Var}_2$ as $\mathrm{Var}[\frac{1}{L+1} \sum_{i=t}^{t+L} Y_i \mid \mathcal{A}_{t:t+L}]$ under $\mathcal{A}_{t:t+L}$ corresponding to Alg. 1 and Alg. 2, respectively. Then the following holds:*

$$\frac{\mathrm{Var}_1}{\mathrm{Var}_0} = \frac{1}{L+1}, \quad \text{and} \quad \frac{\mathrm{Var}_2}{\mathrm{Var}_0} = \frac{1}{(L+1)^2}.$$

Thm. 3.5 shows that the variance of the decision objective reduces when $L$ increases, leading to improved $\mathbb{P}_{\mathrm{AUF}}$ as expectations are the same ($\triangleq \mathbf{s}$ as seen in Appx. C.4). Moreover, the alteration sequence selected by Alg. 2 yields a more substantial variance reduction than that of Alg. 1, resulting in a higher $\mathbb{P}_{\mathrm{AUF}}$ of a same $L$. This phenomenon is demonstrated in Fig. 4, and the intuition is that the variance of $\bar{Y}_{t:t+L} = \frac{1}{L+1} \sum_{i=t}^{t+L} Y_i$ under the rehearsal learning process can be decomposed as:

$$\frac{1}{(L+1)^2} \sum_{i=t}^{t+L} \mathrm{Var}\left[Y_i \mid \mathcal{A}_{t:t+L}\right] + \frac{2}{(L+1)^2} \sum_{j=t+1}^{t+L} \sum_{i=t}^{j-1} \mathrm{Cov}\left[Y_i, Y_j \mid \mathcal{A}_{t:t+L}\right]. \tag{6}$$

Intuitively, the variance of the aggregation is determined by the two components above. Since the rehearsal learning method in Alg. 2 subtracts a weighted $Y_i$ from region center $\mathbf{s}$ at each round, it introduces additional negative correlations between consecutive outcomes $Y_{i+1}$ and $Y_i$. As a result, an additional negative term appears in the second component of Eq. (6) for the aggregation $\mathcal{A}_{t:t+L}$ generated by Alg. 2, further reducing the overall variance compared to the GMuR approach.

Another important consideration is how decision-making is affected when the true parameter set $\boldsymbol{\Theta}$ is unknown, and an estimate $\hat{\boldsymbol{\Theta}}$ is used to approximate matrices $\mathbf{M}, \mathbf{N}, \mathbf{H}$ in QP reformulation Eq. (5).

**Theorem 3.6.** *When $N$ samples are used to estimate $\hat{\boldsymbol{\Theta}}$ as in Appx. C.5, let $\tilde{\mathbf{z}}_t^{\hat{\boldsymbol{\Theta}}}$ denote the alteration selected by solving Eq. (5) with $\hat{\boldsymbol{\Theta}}$, and $\tilde{\mathbf{z}}_t^{\star}$ denote the one selected with true $\boldsymbol{\Theta}$. Under additional assumptions that $\boldsymbol{\Theta}$ is bounded and noise is Gaussian, it holds that $\|\tilde{\mathbf{z}}_t^{\hat{\boldsymbol{\Theta}}} - \tilde{\mathbf{z}}_t^{\star}\|_2 \leq \mathcal{O}(1/\sqrt{N})$.*

Thm. 3.6 ensures that reliable alterations can be identified by our approach without requiring prior knowledge of the true system parameters $\boldsymbol{\Theta}$. Importantly, the samples used for parameter estimation need not be collected through interactions, but can instead be obtained from historical observations. This contrasts with classical reinforcement learning (RL) settings [12], including both online and offline RL as discussed in Appx. 4. Further discussion of our approach is provided in Appx. A.

# 4 Related work

In this section, we review related work on rehearsal learning and two other related directions:

**Rehearsal Learning.** Rehearsal learning aims to uncover the influence relations among variables in AUF and leverage these relations for decision-making [6, 7, 10, 8, 13]. Specifically, prior rehearsal learning methods typically reformulate the intractable probabilistic AUF optimization into a constraint of the form $\mathbb{P}_{\text{AUF}} \geq \tau$, with theoretical guarantees established under linear structural equations with additive Gaussian noise. Qin et al. [6] propose a mixed-integer linear programming (MILP) approach based on sampling techniques to solve the reformulated AUF problem. Du et al. [7] consider decision costs and potential variations in influence relations, and derive an explicit computational formula for contour lines $\mathbb{P}(\mathbf{Y} \in \mathcal{S} \mid \mathbf{x}, Rh(\mathbf{z}^{\xi})) = \tau$ for any $\tau$, leading to a more efficient algorithm. These two approaches could not theoretically maximize $\mathbb{P}_{\text{AUF}}$. Qin et al. [10] then propose a heuristic method to address AUF problem under potentially nonlinear and non-Gaussian settings, though there is no theoretical guarantee and it underperforms prior rehearsal learning approaches when the underlying system is linear. Du et al. [8] present an optimization-based approach to directly optimize the AUF probability, which achieves the optimal AUF probability under the CARE condition though it only considers single-round AUF and assumes Gaussian noise, a setting to which our method naturally reduces. Tao et al. [13] establish a sequential decision-making approach that formulates the single-round AUF into a sequence of sub-problems and solves them sequentially, considering a different setting from ours. Hence, we do not include the latter three methods in our experimental comparison. Our approaches are theoretically established to achieve higher AUF probabilities than prior methods, even under non-Gaussian noise, while also being more efficient w.r.t. $|\mathbf{V}|$.

**Reinforcement Learning (RL).** RL has achieved notable success across a wide range of decision-making tasks [12]. Classical RL methods [14–16] rely on extensive interactions with the environment, which are infeasible in AUF where interactions are rare [2, 6]. While recent advances in offline and hybrid offline-online RL [17–20] aim to reduce the reliance on online interaction by utilizing pre-collected datasets, they do not match the AUF setting because the reward function could diverge between observational and online data. In contrast, rehearsal learning leverages fine-grained structural information

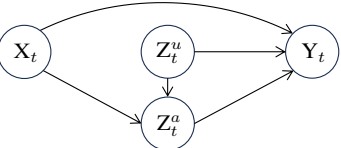

Figure 3: An example illustrating the potential distribution shift.

among variables, enabling effective decision-making from observational data without requiring interactions. To illustrate the challenge faced by RL in this context, consider the example in Fig. 3. Let $X_t$ represent the state, $Z_t^a$ (an actionable variable) the action, and $\mathbb{I}(Y_t \in \mathcal{S})$ the reward. Due to the presence of unactionable variable $Z_t^u$, the conditional distribution $P(Y_t \mid x_t, z_t^a)$ differs from $P(Y_t \mid x_t, Rh(z_t^a))$, where $Rh(z_t^a)$ denotes the result of applying an action based solely on the actionable component. As a result, the reward function inferred from observational data, $\mathbb{I}(Y_t \in \mathcal{S} \mid x_t, z_t^a)$, can deviate significantly from the one with actions, $\mathbb{I}(Y_t \in \mathcal{S} \mid x_t, Rh(z_t^a))$. This discrepancy renders the standard $(s, a, r)$ tuples extracted from observational data ineffective for learning reliable online policies, underscoring the necessity of exploiting structural information in AUF. In addition, a class of model-based RL, namely *causal RL* methods [21–23] incorporates structural information to handle certain types of confounding during offline policy evaluation. However, unlike our approach that explicitly exploits graphical structure (including all ancestral relations), these methods use structure more narrowly and further difficulty in AUF scenarios where the reward ($\bar{Y}_t \in \mathcal{S}$) may vary across rounds. Moreover, rehearsal learning emphasizes identifying the underlying influence relations, which is crucial for human-in-the-loop decision-making. The learned influence relations serves as interpretable guidance for human decision-makers, and since rehearsal itself does not alter the environment, humans can safely choose whether to adopt the suggested decisions; this property lies beyond the scope of RL, which directly interacts with and potentially alters the environment.

**Causality.** Extensive research has explored the use of structural models to support decision-making, most of which is grounded in the framework of SCMs [3]. Some approaches aim at system identification through active interventions or incorporating expert knowledge [24–31], but typically do not consider downstream objectives such as estimating or optimizing causal effects. Additionally, causal bandit methods have been developed to address optimal arm identification problems [32–40]. These approaches generally aim to identify a universally optimal action that maximizes the expected reward. In contrast, rehearsal learning supports mutually influenced relations, and aims to maximize the AUF probability by identifying context-specific optimal actions tailored to different observed contexts $\mathbf{x}$, and allows for a more flexible specification of desired region $\mathcal{S}$ beyond simple maximization.

# 5   Experiments

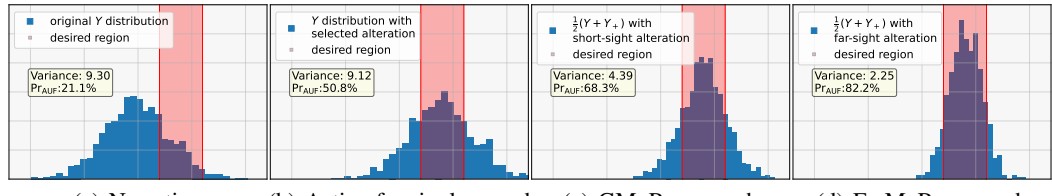

| (a) No action | (b) Action for single-round | (c) GMuR approach | (d) FarMuR approach |

Figure 4: Results on the toy example. $Y$ and $Y_+$ denote values of the target variable in adjacent decision rounds. The results illustrate that variance reduction leads to a significantly higher $\mathbb{P}_{\text{AUF}}$.

We visualize our approaches using a toy example modeling a simplified Texas Hold'em game, followed by evaluations on synthetic and real-world datasets. Our methods are compared against baseline approaches and established rehearsal learning methods, including QWZ23 [6] and MICNS [7]. We do not include RL methods in our comparisons because: (i) they are unsuitable for the AUF setting due to limited interaction opportunities and discrepancies between reward functions in interactional versus observational data, as discussed in Appx. 4; and (ii) previous studies [6, 7] have demonstrated their inadequate performance for AUF tasks. Experimental details are provided in Appx. D.

**Toy example for visualization.** Consider an example involving a simplified Texas Hold'em game, where the variables $\mathbf{X}, \mathbf{Z}, \mathbf{Y}$ are all singletons, and their graphical relations are illustrated in Fig. 5. Let $X_t$ denote the average skill level of the opponents in hand $t$, $Z_t$ the action you take in that hand, and $Y_t$ the yield from hand $t$. Due to overlapping participation across nearby hands, there exist cross-round influences, represented by red edges. Specifically, $X_t \to X_{t+}$ captures the dependence of future opponent skill levels on current

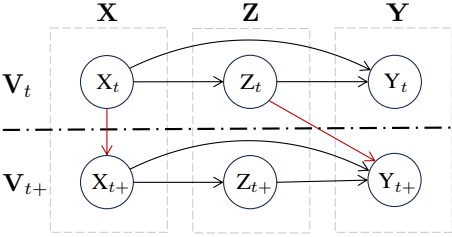

Figure 5: The underlying graphical model

ones, and $Z_t \to Y_{t+}$ reflects the impact of your current action on future yields via revealed playing style. The AUF target is to maintain a moderate average yield over the course of the game. Specifically, suppose we plan to play $L + 1$ hands on the table; let $w = \frac{1}{L+1} \sum_{i=0}^{L} Y_i$ denote the average yield, and we aim to keep it within a safe interval $a \le w \le b$. This is because too low a yield leads to financial loss, while too high a yield may cause potential disputes. Assume that the variables $X_t$ and $Z_t$ are fully observable and rationally controllable, and the noise term is Gaussian.

As illustrated in Fig. 4, if no policy is applied, the actions are passively influenced by $X_t$, resulting in an AUF probability $\mathbb{P}(a \le w \le b)$ of only 21%, as shown in Fig. 4(a). When we actively perform alterations on $Z_t$, the AUF probability increases to 51% for a single-hand outcome, as demonstrated in Fig. 4(b), representing the optimal performance guaranteed by Thm. 3.4. For sequential play (with $L = 1$), the results obtained using the short-sighted and far-sighted policies are shown in Fig. 4(c) and 4(d), respectively. These results demonstrate that our proposed far-sighted FarMuR approach outperforms the short-sighted GMuR approach by better controlling the variance. It is worth noting that the variance in this example is deliberately chosen for clear demonstration purposes. In real games, the variance should not be so small, which would otherwise lead to a significantly lower AUF probability under distributions of $Y$ or $\bar{Y}$, even within a short sequence of rounds under any policy. In what follows, we first briefly introduce the datasets, with details/graph structures listed in Appx. D.

**Synthetic Data.** We construct a synthetic dataset with dimensions of $\mathbf{X}_t, \mathbf{Z}_t,$ and $\mathbf{Y}_t$ set to 2, 4, and 2, respectively, to evaluate cases involving multi-dimensional outcomes $\mathbf{Y}$. Variables $Z^3$ and $Z^4$ are designed to exhibit mutual influence, such that changes in either affect the other. The desired region for target $\bar{\mathbf{Y}}$ is defined as a circular region $\mathcal{S} = \{\mathbf{y} \mid \|\mathbf{y} - \mathbf{s}\|_2 \le 0.8\}$. The parameters of the structural equations, as well as the variances of the additive noise terms, are manually specified.

**Bermuda Data.** The Bermuda dataset records environmental variables in the Bermuda area and has been widely used in prior research [6, 7, 41, 42]. The generation order of variables in this dataset is recorded [43]. The dimensions of $\mathbf{X}_t, \mathbf{Z}_t,$ and $\mathbf{Y}_t$ are 3, 7, and 1, respectively, and the desired region $\mathcal{S}$ for $\bar{\mathbf{Y}}$ is defined as $\mathcal{S} = \{\text{NEC} \in [1.9, 2.0]\}$. The parameters of the structural equations are estimated by fitting least-squares linear models to the normalized data, and the additive noise terms are modeled by residuals using either Gaussian or Laplace distributions.

Table 1: AUF probability $\mathbb{P}(\bar{\mathbf{Y}} \in \mathcal{S} \mid \mathcal{A}_{t:t+L})$ (%) evaluated on two datasets, where $L$ denotes the horizon length $t_e - t_0$. Each value is estimated using 1000 Monte Carlo samples, averaged over 5 random seeds. Results are reported as mean $\pm$ standard deviation, best results are highlighted in bold.

| Dataset | Win. Len. $L$ | No action | QWZ23 [6] | MICNS [7] | Ours (GMuR) | Ours (FarMuR) |
|---|---|---|---|---|---|---|
| Synthetic (Gaussian) | $L=0$ | $0.9 \pm 0.19$ | $4.5 \pm 0.46$ | $\mathbf{4.9 \pm 0.46}$ | $\mathbf{4.9 \pm 0.32}$ | $\mathbf{4.9 \pm 0.32}$ |
| | $L=4$ | $3.7 \pm 0.82$ | $20.3 \pm 0.64$ | $20.1 \pm 0.97$ | $20.7 \pm 0.72$ | $\mathbf{65.1 \pm 0.94}$ |
| | $L=8$ | $3.5 \pm 0.72$ | $32.2 \pm 1.12$ | $31.3 \pm 0.75$ | $32.4 \pm 0.33$ | $\mathbf{95.0 \pm 0.42}$ |
| Synthetic (Laplace) | $L=0$ | $1.0 \pm 0.19$ | $8.4 \pm 0.72$ | $9.9 \pm 0.82$ | $\mathbf{10.2 \pm 0.65}$ | $\mathbf{10.2 \pm 0.65}$ |
| | $L=4$ | $3.9 \pm 0.57$ | $21.3 \pm 0.87$ | $21.5 \pm 1.10$ | $21.9 \pm 0.31$ | $\mathbf{72.8 \pm 1.51}$ |
| | $L=8$ | $3.8 \pm 0.37$ | $33.4 \pm 0.72$ | $33.8 \pm 1.60$ | $34.6 \pm 1.35$ | $\mathbf{93.2 \pm 0.31}$ |
| Bermuda (Gaussian) | $L=0$ | $1.8 \pm 0.26$ | $7.3 \pm 1.19$ | $7.2 \pm 0.47$ | $\mathbf{10.3 \pm 0.85}$ | $\mathbf{10.3 \pm 0.85}$ |
| | $L=4$ | $0.9 \pm 0.29$ | $15.5 \pm 1.34$ | $15.9 \pm 1.19$ | $21.9 \pm 0.26$ | $\mathbf{47.9 \pm 1.29}$ |
| | $L=8$ | $0.4 \pm 0.13$ | $20.8 \pm 1.75$ | $20.2 \pm 2.78$ | $28.2 \pm 1.37$ | $\mathbf{74.3 \pm 1.77}$ |
| Bermuda (Laplace) | $L=0$ | $1.6 \pm 0.15$ | $7.9 \pm 0.87$ | $7.5 \pm 0.67$ | $\mathbf{14.1 \pm 1.29}$ | $\mathbf{14.1 \pm 1.29}$ |
| | $L=4$ | $0.8 \pm 0.27$ | $17.6 \pm 1.31$ | $15.1 \pm 1.64$ | $23.8 \pm 1.19$ | $\mathbf{59.4 \pm 0.94}$ |
| | $L=8$ | $0.4 \pm 0.12$ | $24.1 \pm 0.77$ | $21.5 \pm 1.61$ | $30.1 \pm 1.71$ | $\mathbf{80.3 \pm 1.03}$ |

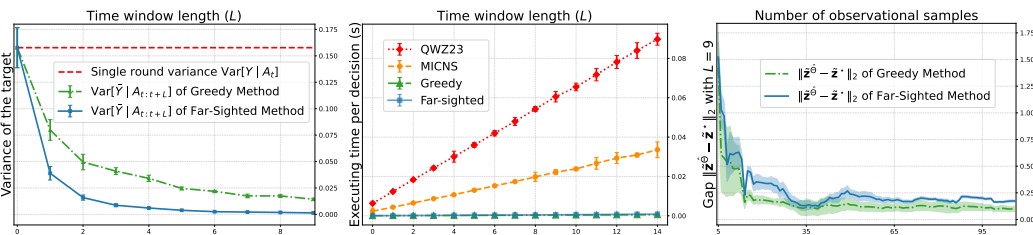

Figure 6: Results on the Bermuda dataset, illustrating the variance reduction w.r.t. the time horizon $L$, average executing time w.r.t. $L$, and gap $\|\tilde{\mathbf{z}}_t^{\hat{\Theta}} - \tilde{\mathbf{z}}_t^{\star}\|_2$ w.r.t. the number of observational samples $N$.

The full results on the two datasets are presented in Tab. 1, demonstrating that our methods outperform existing rehearsal learning approaches as well as the no-action baseline (reflecting the distribution without alteration), under both Gaussian and Laplace noise. Specifically, the GMuR approach Alg. 1 dominates existing methods, as guaranteed by Thm. 3.4, while the FarMuR approach Alg. 2 further improves the AUF probability by leveraging stronger variance reduction, as shown in Thm. 3.5.

Fig. 6 presents additional results on the Bermuda dataset, confirming that the FarMuR method achieves a higher variance reduction rate (left panel, cf. Thm. 3.5) and demonstrates the convergence of excess risk for both our approaches (right panel, cf. Thm. 3.6). The execution time is plotted against $L$ (middle panel), thus the slopes of the lines represent the time complexity w.r.t. other factors beyond $L$, such as $|\mathbf{V}|$, and our methods are significantly more efficient than existing approaches. Although the difference in slopes between our two methods is not prominent in Fig. 6 (as the scale is dominated by the much larger execution times of prior methods), Fig. 10 clarifies that the FarMuR method exhibits a steeper slope than the GMuR method. This highlights a trade-off: as $L$ increases, the FarMuR method offers superior AUF performance but at the cost of increased execution time compared to the GMuR approach. Additional results are provided in Appx. D due to space constraints.

## 6  Conclusion

In this work, we consider an essential class of decision-making tasks termed AUF. Recognizing that practical AUF often hinges on outcomes over extended time horizons, we generalize the rehearsal learning framework and propose an generalized AUF problem formulation to better accommodate long-term scenarios. To address the inherent challenges stemming from the probabilistic nature of the formulated problem, we introduce a QP reformulation and establish its optimality under mild conditions, even with non-Gaussian noise, thereby enhancing the framework's practical utility. Building on the QP reformulation, we develop two novel rehearsal-based algorithms that significantly outperform existing methods. Furthermore, we provide theoretical guarantees for our approach, including variance reduction properties and an excess risk bound when using estimated structural parameters. Experimental results validate both the effectiveness and efficiency of our methods.

## Acknowledgements

This research was supported by NSFC (62495092,62406137), Jiangsu Science Foundation Leading-edge Technology Program (BK20232003), and Collaborative Innovation Center of Novel Software Technology and Industrialization.

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

# A Discussion of our approach

In this section, we discuss the proposed approaches in this work, including the QP reformulation and the two proposed algorithms. First, although the heuristic idea defined in Eq. (5) yields a tractable QP reformulation, it is not always optimal. When assumptions as defined in Ass. 3.3 do not hold, the QP reformulation can be far from optimal. In what follows, we present a counterexample in Fig. 7 that violates the unique target assumption in Ass. 3.3 thus not optimal.

**A counterexample.** Consider a simple example where $T = 0$, $|\mathbf{Y}| = 2$, and $|\mathbf{Z}^\xi| = 1$, with the underlying structural equations defined in Eq. (1) are linear Gaussian. In this case, let $\boldsymbol{\mu} = \mathbb{E}[\mathbf{Y} \mid D, Rh(\mathbf{z}^\xi)]$; it has been proven that $\boldsymbol{\mu} \triangleq [b_1 z^\xi + c_1, b_2 z^\xi + c_2]$ as shown in Eq. (4), where $c_i$ and $b_i$ are constants [6]. Hence, there always exist fixed constants $\alpha$, $\beta$ and $c$ such that $[\alpha, \beta]\boldsymbol{\mu}^\mathsf{T} \triangleq c$, indicating that the trajectory of $\boldsymbol{\mu}$ can be represented as a sloped line according to different values of $z^\xi$, as illustrated in Fig. 7.

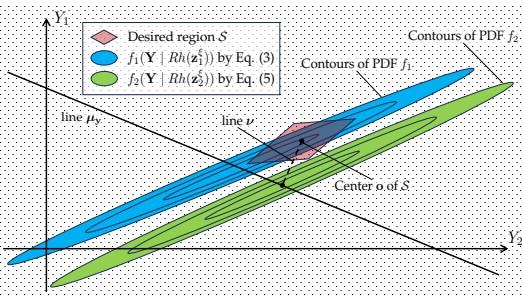

Figure 7: A counterexample of QP reformulation.

Let the diamond region shaded in red denote the desired region $\mathcal{S}$, and draw the vertical line $\boldsymbol{\nu}$ from center $\mathbf{s}$ of $\mathcal{S}$; the intersection point between $\boldsymbol{\nu}$ and the trajectory of $\boldsymbol{\mu}$ corresponds to the solution to Eq. (5), and the probability $\mathbb{P}(\mathbf{Y} \in \mathcal{S} \mid \mathbf{x}, Rh(\mathbf{z}^\xi))$ equals the overlapping area of its probability density function (PDF) and the region $\mathcal{S}$. It is evident that this approach does not maximize the AUF probability, as the intersection point is far from the solution to the original Eq. (3), which is the center of the ellipses shaded in blue. In this case, the center $\boldsymbol{\mu}_\mathbf{y}(\mathbf{z}^\xi)$ of the target variables $\mathbf{Y}$ can only moving within a sub-dimentional space of the target space $\mathbb{R}^{|\mathbf{y}|}$, thus, even when the domin of $\mathbf{z}^\xi$ can take the whole space, $\Delta(\mathbf{Z}) = \mathbb{R}^{|\mathbf{z}^\xi|}$, it not necessarily contains a point $\mathbf{z}^\xi$ such that $\boldsymbol{\mu}_\mathbf{y}(\mathbf{z}^\xi)$ matches $\mathbf{s}$.

Furthermore, we emphasize that our proposed methods primarily address scenarios with fixed start time $t_0$ and end time $t_e$. Nevertheless, Thm. 3.4 guarantees that the QP reformulation remains optimal (conditioned on existing information) in any current decision round for arbitrary $t$ and $T$. Consequently, our approaches can be readily extended to scenarios with varying time horizons, such as when $t_e$ is not fixed but instead maintains a constant distance $L$ from the current time $t$.

# B Definition

In this section, we provide comprehensive definitions and discussions of the Structural Rehearsal Model (SRM), a probabilistic graphical model introduced by [6] to represent influence relations among variables in AUF problem. The SRM comprises a set of rehearsal graphs and associated parameters (for the structural equations) $\{\langle G_t, \boldsymbol{\theta}_t \rangle\}$. The original definitions and discussions of the SRM can be found in Qin et al. [6].

The rehearsal graph $G_t$ models the qualitative influence relations among variables, which is denoted by $G_t = (\mathbf{V}_t, \mathbf{E}_t)$. Specifically, $\mathbf{V}_t$ represents the variable set of the AUF problem and $\mathbf{E}_t$ represents the edges expressing influence relations among variables in round $t$. There are two types of edges in $G_t$, a directional edge $X \to Y$ means that $X$ influences $Y$, and a bi-directional edge $X \leftrightarrow Y$ means that $X$ and $Y$ are mutually influenced. For example, sunlight unilaterally influences the plant growth, whereas rainfall and river flow are mutually influenced, as changes in either one affect the other. The definition of the rehearsal graph is as follows.

**Definition B.1** (Mixed graph, [6]). Let $G = (\mathbf{V}, \mathbf{E})$ be a graph, where $\mathbf{V}$ denotes the vertices and $\mathbf{E}$ the edges. $G$ is a mixed graph if for any distinct vertices $u, v \in \mathbf{V}$, there is at most one edge connecting them, and the edge is either directional ($u \to v$ or $u \leftarrow v$) or bi-directional ($u \leftrightarrow v$).

**Definition B.2** (Bi-directional clique, [6]). A bi-directional clique $C = (\mathbf{V}^c, \mathbf{E}^c)$ of a mixed graph $G = (\mathbf{V}, \mathbf{E})$ is a complete subgraph induced by $\mathbf{V}^c \subseteq \mathbf{V}$ such that any edge $e \in \mathbf{E}^c$ is bi-directional. $C$ is maximal if adding any other vertex does not induce a bi-directional clique.

**Definition B.3** (Rehearsal graph, [6]). Let $G = (\mathbf{V}, \mathbf{E})$ be a mixed graph. Let $\{C_i\}_{i=1}^l$ denote all maximal bi-directional cliques of $G$, where $C_i = (\mathbf{V}_i^c, \mathbf{E}_i^c)$. $G$ is a rehearsal graph if and only if:

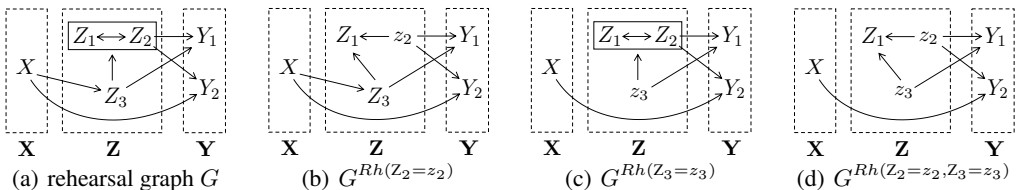

| (a) rehearsal graph $G$ | (b) $G^{Rh(Z_2=z_2)}$ | (c) $G^{Rh(Z_3=z_3)}$ | (d) $G^{Rh(Z_2=z_2, Z_3=z_3)}$ |

Figure 8: An example reproduced from Du et al. [7]. Fig. 8(a) is a rehearsal graph, Fig. 8(b)~8(d) show alteration graphs under different alterations. When an alteration is applied to certain variables, all incoming arrows to these variables are removed, while the rest structure remains unchanged.

1. $\mathbf{V}_i^c \cap \mathbf{V}_j^c = \emptyset$ for any $i \neq j$.
2. $\forall i \in [l]$, if there is any edge pointing from some $u \in \mathbf{V} \backslash \mathbf{V}_i^c$ to some $v \in \mathbf{V}_i^c$, then $\forall v \in \mathbf{V}_i^c$, $u \to v$.
3. There exists a topological ordering for $\{C_i\}_{i=1}^l$ following the directions of directional edges between $C_i$s.

The topological order of bi-directional cliques $\{C_i\}_{i=1}^l$ reflects the generation order. We generalize the definition to time series cases ($\mathbf{G}_K$), with the definition of structural equations as in Sec. 2.

Besides, the decision-making process focuses on identifying suitable *alterations*, as defined in Eq. (3) and Eq. (5). An alteration $\xi$ refers to a decision action specified by human decision-makers, represented as a set of vertex-value pairs (e.g., $\xi \leftarrow \{Z_1 = z_1\}$ in Fig. 8(b)). Meanwhile, the *rehearsal operation*, denoted by $Rh(\cdot)$, corresponds to executing a given alteration, thereby modifying the original graph structure as illustrated in Fig. 8(b)–Fig. 8(d). Specifically, the rehearsal operation removes all original influence links that point into vertices involved in $\xi$, and fixes the values of these vertices according to $\xi$; meanwhile, it preserves the influence relations among the remaining vertices in the resulting graph $G^{Rh(\xi)}$. In the time series setting where alterations are performed over the current and the following $T$ time steps, the operation is denoted by $Rh(\mathbf{z}_t^\xi, \mathbf{z}_{t+1}^\xi, \ldots, \mathbf{z}_{t+T}^\xi)$ to represent this sequential process, the same as Eq. (3) and Eq. (5).

## C    Proofs

### C.1    Proof of Prop. 3.1

**Proposition 3.1.** *Let $\tilde{\mathbf{z}}_t^\xi$ and $\tilde{\mathbf{e}}_t$ denote $\left[\mathbf{z}_{t+T}^{\xi\ \mathsf{T}}, \mathbf{z}_{t+T-1}^{\xi\ \mathsf{T}}, \cdots, \mathbf{z}_t^{\xi\ \mathsf{T}}\right]^\mathsf{T}$ and $\left[\boldsymbol{\varepsilon}_{t+T}^\mathsf{T}, \boldsymbol{\varepsilon}_{t+T-1}^\mathsf{T}, \cdots, \boldsymbol{\varepsilon}_t^\mathsf{T}\right]^\mathsf{T}$. Given $\boldsymbol{\Theta}$ ($\supseteq \mathbf{A}, \mathbf{B}$ in Eq. (2)), $D_t$ (including $\mathbf{v}_{t-1}$ and $\mathbf{x}_t$) and alteration sequence $\tilde{\mathbf{z}}_t^\xi$, it holds that:*

$$\frac{1}{T+1} \sum_{i=t}^{t+T} \mathbf{Y}_i = \mathbf{M}\mathbf{x}_t + \mathbf{N}\mathbf{v}_{t-1} + \mathbf{H}\tilde{\mathbf{z}}_t^\xi + \mathbf{F}\tilde{\mathbf{e}}_t,$$

*where $\mathbf{M} \in \mathbb{R}^{|\mathbf{Y}| \times |\mathbf{X}|}, \mathbf{N} \in \mathbb{R}^{|\mathbf{Y}| \times |\mathbf{V}|}, \mathbf{H} \in \mathbb{R}^{|\mathbf{Y}| \times (T+1)|\mathbf{Z}|}$ and $\mathbf{F} \in \mathbb{R}^{|\mathbf{Y}| \times (T+1)|\mathbf{V}|}$ are constant matrices based on parameters $\boldsymbol{\Theta}$ and time period $T$, while $\boldsymbol{\varepsilon}_t, \cdots, \boldsymbol{\varepsilon}_{t+T}$ are i.i.d. noise vectors.*

*Proof.* Let $\mathbf{E_x} \triangleq (\mathbf{I}_{x \times x}, \mathbf{0}_{x \times z}, \mathbf{0}_{x \times y})^\mathsf{T}, \mathbf{E_z} \triangleq (\mathbf{0}_{z \times x}, \mathbf{I}_{z \times z}, \mathbf{0}_{z \times y})^\mathsf{T}, \mathbf{E_y} \triangleq (\mathbf{0}_{y \times x}, \mathbf{0}_{y \times z}, \mathbf{I}_{y \times y})^\mathsf{T}$, and let $\mathbf{V}_t = [\mathbf{X}_t, \mathbf{Z}_t, \mathbf{Y}_t]$ denote the variable list. Under natural process, it holds that:
$$\mathbf{V}_t = \mathbf{A}\mathbf{V}_t + \mathbf{B}\mathbf{V}_{t-1} + \boldsymbol{\varepsilon}_t.$$

Then under control sequence without observing $\mathbf{x}$, it holds that:
$$\mathbf{V}_t \mid Rh(\mathbf{z}_t^\xi) := \mathbf{E_z}\mathbf{z}_t^\xi + \left(\mathbf{E_x}\mathbf{E_x}^\mathsf{T} + \mathbf{E_y}\mathbf{E_y}^\mathsf{T}\right)(\mathbf{A}\mathbf{V}_t + \mathbf{B}\mathbf{V}_{t-1} + \boldsymbol{\varepsilon}_t),$$

If we can observe $\mathbf{X}_t = \mathbf{x}_t$ under control sequence, then it holds that:
$$\mathbf{V}_t \mid \mathbf{x}_t, Rh(\mathbf{z}_t^\xi) := \mathbf{E_x}\mathbf{x}_t + \mathbf{E_z}\mathbf{z}_t^\xi + \left(\mathbf{E_y}\mathbf{E_y}^\mathsf{T}\right)(\mathbf{A}\mathbf{V}_t + \mathbf{B}\mathbf{V}_{t-1} + \boldsymbol{\varepsilon}_t),$$

Starting from time point just observed $\mathbf{X}_k = \mathbf{x}_k$, it holds that (omiting "$\mid D_t, Rh(\mathbf{z}_t^\xi, \cdots, \mathbf{z}_{t+T}^\xi)$"):
$$\mathbf{V}_{k+i} = \begin{cases} \mathbf{E_z}\mathbf{z}_{k+i}^\xi + \left(\mathbf{E_x}\mathbf{E_x}^\mathsf{T} + \mathbf{E_y}\mathbf{E_y}^\mathsf{T}\right)(\mathbf{A}\mathbf{V}_{k+i} + \mathbf{B}\mathbf{V}_{k+i-1} + \boldsymbol{\varepsilon}_{k+i}) & i \geq 1, \\ \mathbf{E_x}\mathbf{x}_k + \mathbf{E_z}\mathbf{z}_k^\xi + \left(\mathbf{E_y}\mathbf{E_y}^\mathsf{T}\right)(\mathbf{A}\mathbf{V}_k + \mathbf{B}\mathbf{v}_{k-1} + \boldsymbol{\varepsilon}_k) & i = 0. \end{cases} \tag{7}$$

Considering that only $\mathbf{v}_{k-1}$ and $\mathbf{x}_k$ are observed, we want to express $\mathbf{V}_{k+i}$ by $\mathbf{v}_{k-1}$, $\mathbf{x}_k$, and $\{\mathbf{z}_k^\xi, \cdots, \}$.

Let $\mathbf{U} = \left(\mathbf{I} - \left(\mathbf{E_x E_x^\mathsf{T}} + \mathbf{E_y E_y^\mathsf{T}}\right)\mathbf{A}\right)^{-1}\mathbf{E_z}$, $\mathbf{C} = \left(\mathbf{I} - \left(\mathbf{E_x E_x^\mathsf{T}} + \mathbf{E_y E_y^\mathsf{T}}\right)\mathbf{A}\right)^{-1}\left(\mathbf{E_x E_x^\mathsf{T}} + \mathbf{E_y E_y^\mathsf{T}}\right)$, and $\mathbf{\Gamma} = \mathbf{CB}$, it can be derived that (it can be verified that $\mathbf{E_z^\mathsf{T} V}_{k+i} \triangleq \mathbf{z}_{k+i}^\xi$):

$$\mathbf{V}_{k+i} = \mathbf{U}\mathbf{z}_{k+i}^\xi + \mathbf{\Gamma V}_{k+i-1} + \mathbf{C}\varepsilon_{k+i}, \quad i \geq 1.$$

Meanwhile, let $\tilde{\mathbf{U}}$ denote $\left(\mathbf{I} - \mathbf{E_y E_y^\mathsf{T} A}\right)^{-1}\mathbf{E_z}$, $\tilde{\mathbf{C}}$ denote $\left(\mathbf{I} - \mathbf{E_y E_y^\mathsf{T} A}\right)^{-1}\mathbf{E_y E_y^\mathsf{T}}$, $\tilde{\mathbf{\Gamma}}$ denote $\tilde{\mathbf{C}}\mathbf{B}$, and $\mathbf{\Xi}$ denote $\left(\mathbf{I} - \mathbf{E_y E_y^\mathsf{T} A}\right)^{-1}\mathbf{E_x}$ (it can be verified that $\mathbf{E_z^\mathsf{T} V}_k \triangleq \mathbf{z}_k^\xi$ and $\mathbf{E_x^\mathsf{T} V}_k \triangleq \mathbf{x}_k$):

$$\mathbf{V}_k = \mathbf{\Xi x}_k + \tilde{\mathbf{U}}\mathbf{z}_k^\xi + \tilde{\mathbf{\Gamma}}\mathbf{v}_{k-1} + \tilde{\mathbf{C}}\varepsilon_k \tag{8}$$

Iteratively using the equations above, it can be derived for $i \geq 1$ that:

$$
\begin{aligned}
\mathbf{V}_{k+i} =& \mathbf{\Gamma}^i \mathbf{V}_k + \sum_{j=0}^{i-1}\mathbf{\Gamma}^j\left(\mathbf{U}\mathbf{z}_{k+i-j}^\xi + \mathbf{C}\varepsilon_{k+i-j}\right) \\
=& \mathbf{\Gamma}^i\left(\mathbf{\Xi x}_k + \tilde{\mathbf{U}}\mathbf{z}_k^\xi + \tilde{\mathbf{\Gamma}}\mathbf{v}_{k-1} + \tilde{\mathbf{C}}\varepsilon_k\right) + \sum_{j=0}^{i-1}\mathbf{\Gamma}^j\left(\mathbf{U}\mathbf{z}_{k+i-j}^\xi + \mathbf{C}\varepsilon_{k+i-j}\right) \\
=& \underbrace{\mathbf{\Gamma}^i\left(\mathbf{\Xi x}_k + \tilde{\mathbf{\Gamma}}\mathbf{v}_{k-1}\right)}_{\text{constant}} + \underbrace{\left(\mathbf{\Gamma}^i\tilde{\mathbf{U}}\mathbf{z}_k^\xi + \sum_{j=0}^{i-1}\mathbf{\Gamma}^j\mathbf{U}\mathbf{z}_{k+i-j}^\xi\right)}_{\text{affect of control variables}} + \underbrace{\left(\mathbf{\Gamma}^i\tilde{\mathbf{C}}\varepsilon_k + \sum_{j=0}^{i-1}\mathbf{\Gamma}^j\mathbf{C}\varepsilon_{k+i-j}\right)}_{\text{affect of random noises}}
\end{aligned}
$$

Clearly, the value of $\mathbf{Y}_{k+i}$, *i.e.*, $\mathbf{E_y^\mathsf{T} V}_{k+i}$, is affected by the sequence of control variable $\{\mathbf{z}_{t+i}\}_{i=0}^p$, and the more closer, the influence will be more big, because the invertible series limits that $\|\mathbf{\Gamma}\|_2 < 1$.

Meanwhile, noticing that $\sum_{i=0}^T \mathbf{Y}_{t+i} = \mathbf{E_y^\mathsf{T}}\sum_{i=0}^T \mathbf{V}_{t+i}$, thus $\sum_{i=t}^{t+T}\mathbf{Y}_i$ can be expressed as:

$$
\mathbf{E_y^\mathsf{T}}\sum_{i=0}^T\mathbf{\Gamma}^i\left(\mathbf{\Xi x}_t + \tilde{\mathbf{\Gamma}}\mathbf{v}_{t-1}\right) + \mathbf{E_y^\mathsf{T}}\sum_{i=1}^T\sum_{j=0}^{i-1}\mathbf{\Gamma}^j\left(\mathbf{U}\mathbf{z}_{t+i-j}^\xi + \mathbf{C}\varepsilon_{t+i-j}\right) + \mathbf{E_y^\mathsf{T}}\sum_{i=0}^T\mathbf{\Gamma}^i\left(\tilde{\mathbf{U}}\mathbf{z}_t^\xi + \tilde{\mathbf{C}}\varepsilon_t\right)
$$

$$
= \left(\mathbf{E_y^\mathsf{T}}\sum_{i=0}^T\mathbf{\Gamma}^i\right)\left(\mathbf{\Xi x}_t + \tilde{\mathbf{\Gamma}}\mathbf{v}_{t-1}\right) + \mathbf{E_y^\mathsf{T}}\left[\mathbf{IU}, (\mathbf{I}+\mathbf{\Gamma})\mathbf{U}, \cdots, \sum_{i=0}^{T-1}\mathbf{\Gamma}^i\mathbf{U}, \sum_{i=0}^T\mathbf{\Gamma}^i\tilde{\mathbf{U}}\right]\begin{bmatrix}\mathbf{z}_{t+T}^\xi \\ \mathbf{z}_{t+T-1}^\xi \\ \vdots \\ \mathbf{z}_{t+1}^\xi \\ \mathbf{z}_t^\xi\end{bmatrix}
$$

$$
+ \mathbf{E_y^\mathsf{T}}\left[\mathbf{IC}, (\mathbf{I}+\mathbf{\Gamma})\mathbf{C}, \cdots, \sum_{i=0}^{T-1}\mathbf{\Gamma}^i\mathbf{C}, \sum_{i=0}^T\mathbf{\Gamma}^i\tilde{\mathbf{C}}\right]\begin{bmatrix}\varepsilon_{t+T} \\ \varepsilon_{t+T-1} \\ \vdots \\ \varepsilon_{t+1} \\ \varepsilon_t\end{bmatrix}
\tag{9}
$$

In this case, let $\mathbf{M} = \frac{1}{T+1}\mathbf{E_y^\mathsf{T}}\sum_{i=0}^T\mathbf{\Gamma}^i\mathbf{\Xi}$, $\mathbf{H} = \frac{1}{T+1}\mathbf{E_y^\mathsf{T}}\left[\mathbf{IU}, (\mathbf{I}+\mathbf{\Gamma})\mathbf{U}, \cdots, \sum_{i=0}^{T-1}\mathbf{\Gamma}^i\mathbf{U}, \sum_{i=0}^T\mathbf{\Gamma}^i\tilde{\mathbf{U}}\right]$, $\mathbf{N} = \frac{1}{T+1}\mathbf{E_y^\mathsf{T}}\sum_{i=0}^T\mathbf{\Gamma}^i\tilde{\mathbf{\Gamma}}$, $\mathbf{F} = \frac{1}{T+1}\mathbf{E_y^\mathsf{T}}\left[\mathbf{IC}, (\mathbf{I}+\mathbf{\Gamma})\mathbf{C}, \cdots, \sum_{i=0}^{T-1}\mathbf{\Gamma}^i\mathbf{C}, \sum_{i=0}^T\mathbf{\Gamma}^i\tilde{\mathbf{C}}\right]$, Prop. 3.1 is proven.

$\square$

## C.2 Proof of Prop. 3.2

**Lemma C.1** (Prékopa, 1973, Theorem 7). *Let $f, g$ be logarithmic concave functions defined in the space $\mathbb{R}^p$. Then the convolution of these functions, i.e.,*

$$\int_{\mathbb{R}^p} f(\mathbf{x} - \mathbf{y})g(\mathbf{y})d\mathbf{y},$$

*is also logarithmic concave in the entire space $\mathbb{R}^p$.*

**Lemma C.2** (Dharmadhikari and Joag-Dev, 1988, Lemma 2.1). *Suppose a $p$-dimentional random vector $\mathbf{v}$ is log-concavely distributed, and let $\mathbf{M}$ denote a constant matrix with shapes $\mathbb{R}^{m \times p}$, $m \le p$. Then $\mathbf{Mv}$ is also log-concavely distributed.*

**Lemma C.3.** *Suppose $p$-dimentional random vectors $\mathbf{v}_1, \mathbf{v}_2$ are independent and are log-concavely distributed. Then $\mathbf{v}_1 + \mathbf{v}_2$ is also log-concavely distributed.*

*Proof.* Let $f_1, f_2$ denote the PDF of random vectors $\mathbf{v}_1, \mathbf{v}_2$, and let $f$ denote the PDF of random vector $\mathbf{v} = \mathbf{v}_1 + \mathbf{v}_2$. By definition, it can be derived that:

$$f(\mathbf{v}) = \int_{\mathbb{R}^p} f_{\mathbf{v}_1 \mathbf{v}_2}(\mathbf{v} - \mathbf{v}_1, \mathbf{v}_1)d\mathbf{v}_1$$

$$= \int_{\mathbb{R}^p} f_2(\mathbf{v} - \mathbf{v}_1)f_1(\mathbf{v}_1)d\mathbf{v}_1$$

By Lemma C.1, the PDF $f$ is logarithmic concave, thus $\mathbf{v}_1 + \mathbf{v}_2$ is also log-concavely distributed. □

**Proposition 3.2.** *Let $\mathbf{e} \triangleq \mathbf{F}\tilde{\mathbf{e}}_t$ denote the aggregated noise term as defined in Eq. (4). If $\boldsymbol{\varepsilon}_i$ follows a symmetric log-concavely distribution, it always holds that $\mathbf{e}$ is also log-concavely distributed and symmetric about the origin, for any finite time point $t \in \mathbb{Z}_+$ and finite time window $T \in \mathbb{Z}_+$.*

*Proof.* For the assert of log-concave distribution, it can be proved by induction that:

- For $k = 2$, by Lemma C.2 and Lemma C.3, it can be proven that $\mathbf{F}_1 \boldsymbol{\varepsilon}_{t+1} + \mathbf{F}_2 \boldsymbol{\varepsilon}_t$ also obeys a log-concave distribution, where $\mathbf{F}_i$ is the $i$-th part of $\mathbf{F}$;
- Assuming that for $k = k' (\ge 2)$, it also holds that $\sum_{i=1}^{k'} \mathbf{F}_i \boldsymbol{\varepsilon}_{t+k'-i}$ obeys a log-concave distribution;
- For $k = k' + 1$, it holds that $\sum_{i=1}^{k'+1} \mathbf{F}_i \boldsymbol{\varepsilon}_{t+k'+1-i} = \mathbf{F}_{k'+1} \boldsymbol{\varepsilon}_t + \sum_{i=1}^{k'} \mathbf{F}_i \boldsymbol{\varepsilon}_{t+k'+1-i}$. By Lemma C.2, Lemma C.3 and the inductive hypothesis, $\sum_{i=1}^{k'+1} \mathbf{F}_i \boldsymbol{\varepsilon}_{t+k'+1-i}$ also obeys a log-concave distribution.

For symmetric property, it can be proven that $-\mathbf{e} =_d -\mathbf{F}\tilde{\mathbf{e}}_t =_d \mathbf{F}(-\tilde{\mathbf{e}}_t) =_d \mathbf{F}\tilde{\mathbf{e}}_t =_d \mathbf{e}$, where $=_d$ means obeying the same distribution. The third $=_d$ holds because the noise is assumed to be symmetric in Ass. 3.3. □

## C.3 Proof of Thm. 3.4

**Lemma C.4** (Anderson, 1955). *Let $\boldsymbol{x} \in \mathbb{R}^d$ be a random vector with probability density function $f(\boldsymbol{x})$ such that (i) $f(\boldsymbol{x}) = f(-\boldsymbol{x})$ and (ii) $\{\boldsymbol{y} \mid f(\boldsymbol{y}) \ge u\}$ is convex for every $u(0 \le u < \infty)$. If $\mathcal{P}$ is a convex set on $\mathbb{R}^d$, symmetric about the origin, then let $\boldsymbol{c}$ denote an arbitrary constant vector on $\mathbb{R}^d$, it holds that:*

$$\mathbb{P}(\boldsymbol{x} + k\boldsymbol{c} \in \mathcal{P}) \ge \mathbb{P}(\boldsymbol{x} + \boldsymbol{c} \in \mathcal{P}), 0 \le k \le 1.$$

**Theorem 3.4.** *Let $\Delta(\mathbf{Z}) = \mathbb{R}^{|\mathbf{z}^\xi|}$ and $\tilde{\mathbf{z}}_t^\star$ denote the solution to the QP defined in Eq. (5). Under Ass. 3.3 with any finite $t, T \in \mathbb{Z}_+$, the following inequality holds for any alternative $\tilde{\mathbf{z}}_t^a$:*

$$\mathbb{P}\left(\frac{1}{T+1}\sum_{i=t}^{t+T}\mathbf{Y}_i \in \mathcal{S} \ \middle| \ D_t, Rh(\tilde{\mathbf{z}}_t^\star)\right) \ge \mathbb{P}\left(\frac{1}{T+1}\sum_{i=t}^{t+T}\mathbf{Y}_i \in \mathcal{S} \ \middle| \ D_t, Rh(\tilde{\mathbf{z}}_t^a)\right).$$

*Proof.* Recognizing that when we simultaneously shift the distribution and the desired region, the probability mass will not change, *i.e.*, given $\forall \boldsymbol{v} \in \mathbb{R}^{|\mathbf{Y}|}$, let $\mathcal{S}'$ denote the region of the same shape as $\mathcal{S}$ while centered at $\mathbf{s}' = \mathbf{s} - \boldsymbol{v}$ ($\mathbf{s}$ is the symmetric center of region $\mathcal{S}$), then it always holds that:

$$\mathbb{P}\left(\frac{1}{T+1}\sum_{i=t}^{t+T}\mathbf{Y}_i \in \mathcal{S} \ \Big| \ D_t, Rh(\tilde{\mathbf{z}}_t^\star)\right) \triangleq \mathbb{P}\left(\frac{1}{T+1}\sum_{i=t}^{t+T}\mathbf{Y}_i - \boldsymbol{v} \in \mathcal{S}' \ \Big| \ D_t, Rh(\tilde{\mathbf{z}}_t^\star)\right).$$

Meanwhile, recall from Prop. 3.1 that after alteration $\tilde{\mathbf{z}}_t^\xi$, $\frac{1}{T+1}\sum_{i=t}^{t+T}\mathbf{Y}_i$ can be expressed as:

$$\frac{1}{T+1}\sum_{i=t}^{t+T}\mathbf{Y}_i = \mathbf{M}\mathbf{x}_t + \mathbf{N}\mathbf{v}_{t-1} + \mathbf{H}\tilde{\mathbf{z}}_t^\xi + \mathbf{F}\tilde{\mathbf{e}}_t.$$

Hence, let $\boldsymbol{v} \triangleq \boldsymbol{v}(\tilde{\mathbf{z}}_t^\xi) = \mathbf{M}\mathbf{x}_t + \mathbf{N}\mathbf{v}_{t-1} + \mathbf{H}\tilde{\mathbf{z}}_t^\xi$ and let $\mathbf{e}$ denote $\mathbf{F}\tilde{\mathbf{e}}_t$, it always holds that:

$$\mathbb{P}\left(\frac{1}{T+1}\sum_{i=t}^{t+T}\mathbf{Y}_i \in \mathcal{S} \ \Big| \ D_t, Rh(\tilde{\mathbf{z}}_t^\xi)\right) \triangleq \mathbb{P}\left(\mathbf{e} \in \mathcal{S}' \ \Big| \ D_t, Rh(\tilde{\mathbf{z}}_t^\xi)\right)$$
$$\triangleq \mathbb{P}\left(\mathbf{e} - \mathbf{s}' \in \mathcal{S}_\mathbf{0} \ \Big| \ D_t, Rh(\tilde{\mathbf{z}}_t^\xi)\right),$$

where $\mathcal{S}'$ is the region of the same shape as $\mathcal{S}$ while centered at $\mathbf{s}' = \mathbf{s} - (\mathbf{M}\mathbf{x}_t + \mathbf{N}\mathbf{v}_{t-1} + \mathbf{H}\tilde{\mathbf{z}}_t^\xi)$, and $\mathcal{S}_\mathbf{0}$ is the region of the same shape as $\mathcal{S}$ while centered at the origin $\mathbf{0}$.

If we can prove that: (a) $\mathbf{e}$ is a random vector always satisfying the constraints (i) and (ii) in Lemma C.4; and (b) $\mathbf{s}' = \mathbf{0}$ if $\tilde{\mathbf{z}}_t^\xi = \tilde{\mathbf{z}}_t^\star$, then Thm. 3.4 can be straightforwardly proven according to Lemma C.4.

We first prove (a) that $\mathbf{e}$ is a random vector always satisfying the constraints (i) and (ii) in Lemma C.4. Since it is guaranteed by Prop. 3.2 that $\mathbf{e}$ is always symmetric about the origin for $\forall$ finite time point $t \in \mathbb{Z}_+$ and finite time window $T \in \mathbb{Z}_+$, thus the symmetric property is holds for (i) in Lemma C.4. For (ii) in Lemma C.4, let $f_\mathbf{e}(\cdot)$ denote the PDF of random vector $\mathbf{e}$, it can be derived that for $\forall u \geq 0$:

$$\{\boldsymbol{y} \mid f_\mathbf{e}(\boldsymbol{y}) \geq u\} \Leftrightarrow \{\boldsymbol{y} \mid \log f_\mathbf{e}(\boldsymbol{y}) \geq \log u\}.$$

Meanwhile, following from Prop. 3.2 that $\mathbf{e}$ is always log-concavely distributed for $\forall$ finite time point $t \in \mathbb{Z}_+$ and finite time window $T \in \mathbb{Z}_+$, by definition of log-concave distribution, the logarithmic PDF $\log f_\mathbf{e}(\cdot)$ is concave. Hence, for $\forall \boldsymbol{y}_1, \boldsymbol{y}_2 \in \{\boldsymbol{y} \mid \log f_\mathbf{e}(\boldsymbol{y}) \geq \log u\}$ and $\lambda \in [0, 1]$, it can be derived that:

$$\log f_\mathbf{e}(\lambda \boldsymbol{y}_1 + (1-\lambda)\boldsymbol{y}_2) \geq \lambda \log f_\mathbf{e}(\boldsymbol{y}_1) + (1-\lambda)\log f_\mathbf{e}(\boldsymbol{y}_2)$$
$$\geq \lambda \log u + (1-\lambda)\log u = \log u,$$

*i.e.*, $\lambda \boldsymbol{y}_1 + (1-\lambda)\boldsymbol{y}_2 \in \{\boldsymbol{y} \mid \log f_\mathbf{e}(\boldsymbol{y}) \geq \log u\}$ always holds, illustrating that the set $\{\boldsymbol{y} \mid \log f_\mathbf{e}(\boldsymbol{y}) \geq \log u\}$ is always concave. Because $\log u$ is a reversible function and takses value in $(-\infty, \infty) \supsetneq [0, \infty)$, thus we have proven that (ii) in Lemma C.4 always satisfies for $\mathbf{e}$.

Then, we prove (b) that $\tilde{\mathbf{z}}_t^\xi = \tilde{\mathbf{z}}_t^\star$ can lead to $\mathbf{s}' = \mathbf{0}$. Recall from Eq. (5) that the alteration $\tilde{\mathbf{z}}_t^\star$ is selected by:

$$\tilde{\mathbf{z}}_t^\star = \arg\min_{\tilde{\mathbf{z}}_t^\xi} \ \left\|\mathbf{M}\mathbf{x}_t + \mathbf{N}\mathbf{v}_{t-1} + \mathbf{H}\tilde{\mathbf{z}}_t^\xi - \mathbf{s}\right\|_2.$$

Consider linear equations define as:

$$\mathbf{H}\tilde{\mathbf{z}}_t^\xi = \mathbf{s} - \mathbf{M}\mathbf{x}_t - \mathbf{N}\mathbf{v}_{t-1}, \tag{10}$$

Since $\mathbf{H}$ is row full rank otherwise conflict the **unique target** assumption defined in Ass. 3.3 (and thus the row number is greater or equal than the column number), the augmented matrix $[\mathbf{H} \parallel \mathbf{s} - \mathbf{M}\mathbf{x}_t - \mathbf{N}\mathbf{v}_{t-1}]$ must have the same rank as matrix $\mathbf{H}$. This to say, there at least one solution exist for Eq. (10), and choosing $\tilde{\mathbf{z}}_t^\star$ as the solution to Eq. (10), it can lead to $\mathbf{s}' = \mathbf{s} - (\mathbf{M}\mathbf{x}_t + \mathbf{N}\mathbf{v}_{t-1} + \mathbf{H}\tilde{\mathbf{z}}_t^\xi) \triangleq \mathbf{0}$.

In summary, we have proven that (a) $\mathbf{e}$ is a random vector always satisfying the constraints (i) and (ii) in Lemma C.4; and (b) $\mathbf{s}' = \mathbf{0}$ if $\tilde{\mathbf{z}}_t^\xi = \tilde{\mathbf{z}}_t^\star$; hence, the proof of Thm. 3.4 is established. $\qquad\square$

## C.4 Proof of Thm. 3.5

**Theorem 3.5.** *When $\mathbf{Y}$ is singleton (i.e., $|\mathbf{Y}| = 1$), let $\mathrm{Var}_0 = \mathrm{Var}[Y_t \mid D_t, Rh(\mathbf{z}_t)]$, and let $\mathcal{A}_{t:t+L}$ denote the rehearsal learning process. Define $\mathrm{Var}_1$ and $\mathrm{Var}_2$ as $\mathrm{Var}[\frac{1}{L+1}\sum_{i=t}^{t+L} Y_i \mid \mathcal{A}_{t:t+L}]$ under $\mathcal{A}_{t:t+L}$ corresponding to Alg. 1 and Alg. 2, respectively. Then the following holds:*

$$\frac{\mathrm{Var}_1}{\mathrm{Var}_0} = \frac{1}{L+1}, \quad and \quad \frac{\mathrm{Var}_2}{\mathrm{Var}_0} = \frac{1}{(L+1)^2}.$$

*Proof.* Recall from Eq. (8), conditioned on $Rh(\mathbf{z}_t^{\xi})$ and $D_t$ (including $\mathbf{x}_t$, $\mathbf{v}_{t-1}$), $Y_t$ can be expressed as:

$$Y_t \mid D_t, Rh(\mathbf{z}_t^{\xi}) = \mathbf{e}_{\mathbf{y}}^{\mathsf{T}}\mathbf{V}_t = c_t + \varepsilon_t^y,$$

where $\mathbf{e}_{\mathbf{y}} \triangleq \mathbf{E}_{\mathbf{y}}$ since $\mathbf{E}_{\mathbf{y}}$ is a zero vector with only its last element equal to 1; $c_t$ is a constant depending on $\mathbf{z}_t^{\xi}$, $\mathbf{x}_t$, and $\mathbf{v}_{t-1}$; and $\varepsilon_t^y$ denotes the additive noise associated with the variable $Y_t$, *i.e.*, the last element of $\boldsymbol{\varepsilon}_t$ in Eq. (2). Hence, it follows that:

$$\mathrm{Var}_0 = \langle c_t + \varepsilon_t^y, c_t + \varepsilon_t^y \rangle \triangleq \langle \varepsilon_t^y, \varepsilon_t^y \rangle. \tag{11}$$

To compute $\mathrm{Var}_1$, we review the rehearsal learning process $\mathcal{A}_{t:t+L}$ from the GMuR method Alg. 1. Since Alg. 1 only focuses on the current $Y_i$ in each decision round $i$, $T \triangleq 0$ and it can be derived from Eq. (9) that:

$$\text{In Alg. 1:} \quad \mathbf{M} = \mathbf{e}_{\mathbf{y}}^{\mathsf{T}}\mathbf{\Xi}, \quad \mathbf{N} = \mathbf{e}_{\mathbf{y}}^{\mathsf{T}}\tilde{\boldsymbol{\Gamma}}, \quad \mathbf{H} = \mathbf{e}_{\mathbf{y}}^{\mathsf{T}}\tilde{\mathbf{U}}.$$

Note that $\mathbf{H}$ is a row vector in this case, thus it can be derived that:

$$\mathbf{H}^{\mathsf{T}}(\mathbf{H}\mathbf{H}^{\mathsf{T}})^{-1} = \frac{1}{\mathbf{H}\mathbf{H}^{\mathsf{T}}}\mathbf{H}^{\mathsf{T}},$$

and from line 8 of Alg. 1, it follows that (since $\mathbf{H}$ is a row vector):

$$\mathbf{z}_i^{\xi} = \frac{1}{\mathbf{H}\mathbf{H}^{\mathsf{T}}}\mathbf{H}^{\mathsf{T}}\left(\mathbf{s} - \mathbf{e}_{\mathbf{y}}^{\mathsf{T}}\mathbf{\Xi}\mathbf{x}_i - \mathbf{e}_{\mathbf{y}}^{\mathsf{T}}\tilde{\boldsymbol{\Gamma}}\mathbf{v}_{i-1}\right). \tag{12}$$

Starting from time $t$, it can be derived for $t \le i \le t+L$ that:

$$\begin{aligned}
Y_i &= \mathbf{e}_{\mathbf{y}}^{\mathsf{T}}\left(\mathbf{\Xi}\mathbf{x}_i + \tilde{\mathbf{U}}\mathbf{z}_i^{\xi} + \tilde{\boldsymbol{\Gamma}}\mathbf{v}_{i-1} + \tilde{\mathbf{C}}\boldsymbol{\varepsilon}_i\right) \\
&= \mathbf{e}_{\mathbf{y}}^{\mathsf{T}}\mathbf{\Xi}\mathbf{x}_i + \mathbf{H}\mathbf{z}_i^{\xi} + \mathbf{e}_{\mathbf{y}}^{\mathsf{T}}\tilde{\boldsymbol{\Gamma}}\mathbf{v}_{i-1} + \varepsilon_i^y \quad (\mathbf{e}_{\mathbf{y}}^{\mathsf{T}}\tilde{\mathbf{U}} \triangleq \mathbf{H}) \\
&= \mathbf{e}_{\mathbf{y}}^{\mathsf{T}}\mathbf{\Xi}\mathbf{x}_i + \left(\mathbf{s} - \mathbf{e}_{\mathbf{y}}^{\mathsf{T}}\mathbf{\Xi}\mathbf{x}_i - \mathbf{e}_{\mathbf{y}}^{\mathsf{T}}\tilde{\boldsymbol{\Gamma}}\mathbf{v}_{i-1}\right) + \mathbf{e}_{\mathbf{y}}^{\mathsf{T}}\tilde{\boldsymbol{\Gamma}}\mathbf{v}_{i-1} + \varepsilon_i^y \\
&= \mathbf{s} + \varepsilon_i^y
\end{aligned}$$

The 3rd equality holds from Eq. (12). Hence, it can be computed (under $\mathcal{A}_{t:t+L}$ from Alg. 1) that:

$$\begin{aligned}
\mathrm{Var}_1 &= \left\langle \frac{1}{L+1}\sum_{i=t}^{t+L} Y_i, \frac{1}{L+1}\sum_{i=t}^{t+L} Y_i \right\rangle \\
&= \frac{1}{(L+1)^2}\sum_{i=t}^{t+L} \langle Y_i, Y_i \rangle + \frac{2}{(L+1)^2}\sum_{j=t+1}^{t+L}\sum_{i=t}^{j-1} \langle Y_i, Y_j \rangle \\
&= \frac{1}{(L+1)^2}\sum_{i=t}^{t+L} \langle \mathbf{s} + \varepsilon_i^y, \mathbf{s} + \varepsilon_i^y \rangle + \frac{2}{(L+1)^2}\sum_{j=t+1}^{t+L}\sum_{i=t}^{j-1} \langle \mathbf{s} + \varepsilon_i^y, \mathbf{s} + \varepsilon_j^y \rangle \\
&= \frac{1}{(L+1)^2}\sum_{i=t}^{t+L} \langle \varepsilon_i^y, \varepsilon_i^y \rangle \triangleq \frac{1}{L+1} \langle \varepsilon_t^y, \varepsilon_t^y \rangle
\end{aligned} \tag{13}$$

To compute $\mathrm{Var}_2$, we review the rehearsal learning process $\mathcal{A}_{t:t+L}$ from the FarMuR method Alg. 2. It can be derived from Eq. (9) that in the last round (*i.e.*, $i = t+L$ or $i = t_e$) of Alg. 2:

$$\text{In Alg. 2, decision round } t_e: \quad \mathbf{M} = \frac{1}{L+1}\mathbf{e}_{\mathbf{y}}^{\mathsf{T}}\mathbf{\Xi}, \quad \mathbf{N} = \frac{1}{L+1}\mathbf{e}_{\mathbf{y}}^{\mathsf{T}}\tilde{\boldsymbol{\Gamma}}, \quad \mathbf{H} = \frac{1}{L+1}\mathbf{e}_{\mathbf{y}}^{\mathsf{T}}\tilde{\mathbf{U}}.$$

Note that in round $i = t_e$, the expression of $\mathbf{M}, \mathbf{N}, \mathbf{H}$ are similar to those in Alg. 1 because it is the last decision round. Thus, it can be derived (similar with Eq. (12)) that:

$$\mathbf{z}_{t+L}^{\xi} = \frac{1}{\mathbf{H}\mathbf{H}^{\mathsf{T}}}\mathbf{H}^{\mathsf{T}}\left(\mathbf{s}_{t+L} - \frac{1}{L+1}\mathbf{e}_{\mathbf{y}}^{\mathsf{T}}\mathbf{\Xi}\mathbf{x}_{t+L} - \frac{1}{L+1}\mathbf{e}_{\mathbf{y}}^{\mathsf{T}}\tilde{\mathbf{\Gamma}}\mathbf{v}_{t+L-1}\right). \tag{14}$$

Hence, it follows that:

$$\begin{aligned}
Y_{t+L} &= \mathbf{e}_{\mathbf{y}}^{\mathsf{T}}\left(\mathbf{\Xi}\mathbf{x}_{t+L} + \tilde{\mathbf{U}}\mathbf{z}_{t+L}^{\xi} + \tilde{\mathbf{\Gamma}}\mathbf{v}_{t+L-1} + \tilde{\mathbf{C}}\boldsymbol{\varepsilon}_{t+L}\right) \\
&= \mathbf{e}_{\mathbf{y}}^{\mathsf{T}}\mathbf{\Xi}\mathbf{x}_{t+L} + (L+1)\mathbf{H}\mathbf{z}_{t+L}^{\xi} + \mathbf{e}_{\mathbf{y}}^{\mathsf{T}}\tilde{\mathbf{\Gamma}}\mathbf{v}_{t+L-1} + \varepsilon_{t+L}^{y} \qquad (\mathbf{e}_{\mathbf{y}}^{\mathsf{T}}\tilde{\mathbf{U}} \triangleq (L+1)\mathbf{H}) \\
&= \mathbf{e}_{\mathbf{y}}^{\mathsf{T}}\mathbf{\Xi}\mathbf{x}_{t+L} + \left((L+1)\mathbf{s}_{t+L} - \mathbf{e}_{\mathbf{y}}^{\mathsf{T}}\mathbf{\Xi}\mathbf{x}_{t+L} - \mathbf{e}_{\mathbf{y}}^{\mathsf{T}}\tilde{\mathbf{\Gamma}}\mathbf{v}_{t+L-1}\right) + \mathbf{e}_{\mathbf{y}}^{\mathsf{T}}\tilde{\mathbf{\Gamma}}\mathbf{v}_{t+L-1} + \varepsilon_{t+L}^{y} \\
&= (L+1)\mathbf{s}_{t+L} + \varepsilon_{t+L}^{y} \\
&= \varepsilon_{t+L}^{y} + (L+1)\left(\mathbf{s} - \frac{1}{L+1}\sum_{i=t}^{t+L-1}Y_i\right)
\end{aligned}$$

The 3rd equality holds from Eq. (14) and the 5th equality holds from the updating operation (line 9 of Alg. 2). It can be further derived that:

$$\frac{1}{L+1}\sum_{i=t}^{t+L}Y_i = \mathbf{s} + \frac{1}{L+1}\varepsilon_{t+L}^{y}.$$

Hence, it can be computed (under $\mathcal{A}_{t:t+L}$ from Alg. 2) that:

$$\begin{aligned}
\mathrm{Var}_2 &= \left\langle \frac{1}{L+1}\sum_{i=t}^{t+L}Y_i, \frac{1}{L+1}\sum_{i=t}^{t+L}Y_i \right\rangle \\
&= \left\langle \mathbf{s} + \frac{1}{L+1}\varepsilon_{t+L}^{y}, \mathbf{s} + \frac{1}{L+1}\varepsilon_{t+L}^{y} \right\rangle \\
&= \frac{1}{(L+1)^2}\left\langle \varepsilon_{t+L}^{y}, \varepsilon_{t+L}^{y} \right\rangle \\
&\triangleq \frac{1}{(L+1)^2}\left\langle \varepsilon_{t}^{y}, \varepsilon_{t}^{y} \right\rangle
\end{aligned} \tag{15}$$

Combining Eq. (11), Eq. (13), and Eq. (15) completes the proof of Thm. 3.5.

$\square$

## C.5 Proof of Thm. 3.6

**Theorem 3.6.** *When $N$ samples are used to estimate $\hat{\mathbf{\Theta}}$ as in Appx. C.5, let $\tilde{\mathbf{z}}_t^{\hat{\mathbf{\Theta}}}$ denote the alteration selected by solving Eq. (5) with $\hat{\mathbf{\Theta}}$, and $\tilde{\mathbf{z}}_t^{\star}$ denote the one selected with true $\mathbf{\Theta}$. Under additional assumption that $\mathbf{\Theta}$ is bounded and noise is Gaussian, it holds that $\|\tilde{\mathbf{z}}_t^{\hat{\mathbf{\Theta}}} - \tilde{\mathbf{z}}_t^{\star}\|_2 \leq \mathcal{O}(1/\sqrt{N})$.*

We first present the estimation of $\hat{\mathbf{\Theta}}$ (including $\hat{\mathbf{A}}$ and $\hat{\mathbf{B}}$) as follows. In practical scenarios, the true parameter values are typically unknown. In such cases, we aim to estimate the parameters $\mathbf{A}$ and $\mathbf{B}$ from the historically collected data. Let $\mathbf{P}^j = \begin{bmatrix} \mathrm{PA}_1^j & \cdots & \mathrm{PA}_N^j \end{bmatrix}^{\mathsf{T}}$ denote the matrix of parent values for the $j$-th variable, and $\mathbf{v}^j = \begin{bmatrix} V_1^j, \ldots, V_N^j \end{bmatrix}^{\mathsf{T}}$ the corresponding observed values of the variable itself. Then, the parameters associated with the generation of the $j$-th variable can be estimated via the following least squares estimation (LSE):

$$\arg\min_{\boldsymbol{\beta}^j} \frac{1}{2}\left\|\mathbf{v}^j - \mathbf{P}^j\boldsymbol{\beta}^j\right\|_2^2.$$

Note that the parameter vector $\boldsymbol{\beta}^j$ represents the $j$-th row of the concatenated coefficient matrices $\mathbf{A}$ and $\mathbf{B}$. Then the proof of Thm. 3.6 is detailed as follows.

*Proof.* Define $\boldsymbol{\Omega} \triangleq [\mathbf{A} \quad \mathbf{B}]$. According to Lemma C.3 of Du et al. [7], under the assumptions that $\boldsymbol{\Theta}$ is bounded and the noise follows a Gaussian distribution, the parameter estimation error satisfies:

$$\left\| \hat{\boldsymbol{\Omega}} - \boldsymbol{\Omega} \right\|_F^2 = \left\| \hat{\mathbf{A}} - \mathbf{A} \right\|_F^2 + \left\| \hat{\mathbf{B}} - \mathbf{B} \right\|_F^2 \leq \mathcal{O}\left(\frac{1}{N}\right).$$

We first analyze the Lipschitz continuity of the matrix functions $\mathbf{M}(\boldsymbol{\Omega})$, $\mathbf{N}(\boldsymbol{\Omega})$, and $\mathbf{H}(\boldsymbol{\Omega})$ defined in Eq. (9). Note that $(\mathbf{I} - \mathbf{A})$ is invertible (i.e., full rank), which is Lipschitz continuous w.r.t. $\mathbf{A}$ as proven as: Let $\mathbf{A}_1, \mathbf{A}_2 \in \mathbb{R}^{|\mathbf{V}| \times |\mathbf{V}|}$ be matrices such that $\mathbf{I} - \mathbf{A}_1$ and $\mathbf{I} - \mathbf{A}_2$ are invertible. Let the constant $\gamma > 0$, $\sup_{i=1,2} \left\| (\mathbf{I} - \mathbf{A}_i)^{-1} \right\|_F \leq \gamma$ denote the upper bound of the $F$ norm. It can be derived that:

$$\begin{aligned}
\left\| (\mathbf{I} - \mathbf{A}_1)^{-1} - (\mathbf{I} - \mathbf{A}_2)^{-1} \right\|_F &= \left\| (\mathbf{I} - \mathbf{A}_1)^{-1}(\mathbf{A}_1 - \mathbf{A}_2)(\mathbf{I} - \mathbf{A}_2)^{-1} \right\|_F \\
&\leq \left\| (\mathbf{I} - \mathbf{A}_1)^{-1} \right\|_F \cdot \|\mathbf{A}_1 - \mathbf{A}_2\|_F \cdot \left\| (\mathbf{I} - \mathbf{A}_2)^{-1} \right\|_F \\
&\leq \gamma^2 \|\mathbf{A}_1 - \mathbf{A}_2\|_F.
\end{aligned}$$

In this case the resolvent $(\mathbf{I} - \mathbf{A})^{-1}$ inherits the above Lipschitz property. Besides, the Lipschitz continuity of matrix multiplications (bounded matrices) can also be proven as: For $\forall \boldsymbol{\Omega}_1, \boldsymbol{\Omega}_2$ and bounded matrices $\mathbf{P}(\boldsymbol{\Omega}), \mathbf{Q}(\boldsymbol{\Omega})$ that are Lipschitz continuous w.r.t. $\boldsymbol{\Omega}$ can be multiplicated together, it follows that:

$$\mathbf{P}(\boldsymbol{\Omega}_1)\mathbf{Q}(\boldsymbol{\Omega}_1) - \mathbf{P}(\boldsymbol{\Omega}_2)\mathbf{Q}(\boldsymbol{\Omega}_2) = \mathbf{P}(\boldsymbol{\Omega}_1)\left[\mathbf{Q}(\boldsymbol{\Omega}_1) - \mathbf{Q}(\boldsymbol{\Omega}_2)\right] + \left[\mathbf{P}(\boldsymbol{\Omega}_1) - \mathbf{P}(\boldsymbol{\Omega}_2)\right]\mathbf{Q}(\boldsymbol{\Omega}_2).$$

Applying the triangle inequality for the Frobenius norm, it follows that:

$$\begin{aligned}
&\|\mathbf{P}(\boldsymbol{\Omega}_1)\mathbf{Q}(\boldsymbol{\Omega}_1) - \mathbf{P}(\boldsymbol{\Omega}_2)\mathbf{Q}(\boldsymbol{\Omega}_2)\|_F \\
=& \|\mathbf{P}(\boldsymbol{\Omega}_1)\left[\mathbf{Q}(\boldsymbol{\Omega}_1) - \mathbf{Q}(\boldsymbol{\Omega}_2)\right]\|_F + \|\left[\mathbf{P}(\boldsymbol{\Omega}_1) - \mathbf{P}(\boldsymbol{\Omega}_2)\right]\mathbf{Q}(\boldsymbol{\Omega}_2)\|_F \\
\leq& \|\mathbf{P}(\boldsymbol{\Omega}_1)\|_F\|\mathbf{Q}(\boldsymbol{\Omega}_1) - \mathbf{Q}(\boldsymbol{\Omega}_2)\|_F + \|\mathbf{P}(\boldsymbol{\Omega}_1) - \mathbf{P}(\boldsymbol{\Omega}_2)\|_F\|\mathbf{Q}(\boldsymbol{\Omega}_2)\|_F \\
\leq& U_P\|\mathbf{Q}(\boldsymbol{\Omega}_1) - \mathbf{Q}(\boldsymbol{\Omega}_2)\|_F + U_Q\|\mathbf{P}(\boldsymbol{\Omega}_1) - \mathbf{P}(\boldsymbol{\Omega}_2)\|_F \\
\leq& (U_P L_Q + U_Q L_P)\|\boldsymbol{\Omega}_1 - \boldsymbol{\Omega}_2\|_F.
\end{aligned}$$

$U_P$ and $U_Q$ are finite upper bounds because $\mathbf{P}, \mathbf{Q}$ are bounded matrices.

Since $\mathbf{M}(\boldsymbol{\Omega})$, $\mathbf{N}(\boldsymbol{\Omega})$, and $\mathbf{H}(\boldsymbol{\Omega})$ are constructed via matrix multiplications and such resolvents, they all have Lipschitz continuity w.r.t. $\boldsymbol{\Omega}$. Finally, because $\mathbf{H}$ is row full rank otherwise conflict the **unique target** assumption defined in Ass. 3.3, it can be verified that $\mathbf{H}\mathbf{H}^\mathsf{T}$ is full rank.

Hence, the pseudo-inverse solution $\tilde{\mathbf{z}}_t(\boldsymbol{\Omega}) = \mathbf{H}^\mathsf{T}(\mathbf{H}\mathbf{H}^\mathsf{T})^{-1}(\mathbf{s} - \mathbf{M}\mathbf{x}_t - \mathbf{N}\mathbf{v}_{t-1})$ is Lipschitz continuous w.r.t. $\boldsymbol{\Omega}$ because it has been proven that matrix multiplication and full-rank inverse (similar to the proof of $(\mathbf{I} - \mathbf{A})^{-1}$) preserve Lipschitz continuity. Combining this with the parameter estimation error bound, we conclude:

$$\left\| \tilde{\mathbf{z}}_t^{\hat{\boldsymbol{\Theta}}} - \tilde{\mathbf{z}}_t^\star \right\|_2^2 \triangleq \left\| \tilde{\mathbf{z}}_t(\hat{\boldsymbol{\Omega}}) - \tilde{\mathbf{z}}_t(\boldsymbol{\Omega}) \right\|_2^2 \leq L \left\| \hat{\boldsymbol{\Omega}} - \boldsymbol{\Omega} \right\|_F^2 \leq \mathcal{O}\left(\frac{1}{N}\right).$$

Taking square roots completes the proof: $\left\| \tilde{\mathbf{z}}_t^{\hat{\boldsymbol{\Theta}}} - \tilde{\mathbf{z}}_t^\star \right\|_2 \leq \mathcal{O}(1/\sqrt{N})$. □

# D    Experimental details

We provide the detailed information of Sec. 5 and additional experiments in this section. First, all experiments were run on a Nvidia Tesla A100 GPU and two Intel Xeon Platinum 8358 CPUs. Then we provide true parameters of the synthetic dataset, with variables in the dataset illustrated in Fig. 9.

The synthetic dataset includes $\mathbf{V} = [X^1, X^2, Z^1, Z^2, Z^3, Z^4, Y^1, Y^2]$, and it holds that $\mathbf{V}_t = \mathbf{A}\mathbf{V}_t + \mathbf{B}\mathbf{V}_{t-1} + \boldsymbol{\varepsilon}_t$, where the instantaneous influence relations are recorded in:

$$\mathbf{A} = \begin{bmatrix}
0 & 0 & 0 & 0 & 0 & 0 & 0 & 0 \\
0 & 0 & 0 & 0 & 0 & 0 & 0 & 0 \\
0 & 1 & 0 & 0 & 0 & 0 & 0 & 0 \\
1 & 0 & 0 & 0 & 0 & 0 & 0 & 0 \\
0 & 0 & 0.5 & 1.3 & 0 & 0 & 0 & 0 \\
0 & 0 & 2 & 0.4 & 0 & 0 & 0 & 0 \\
0 & 0 & -1 & 0 & 0 & 0.9 & 0 & 0 \\
0 & 0 & 1.6 & 0 & 0 & -0.5 & 0 & 0
\end{bmatrix},$$

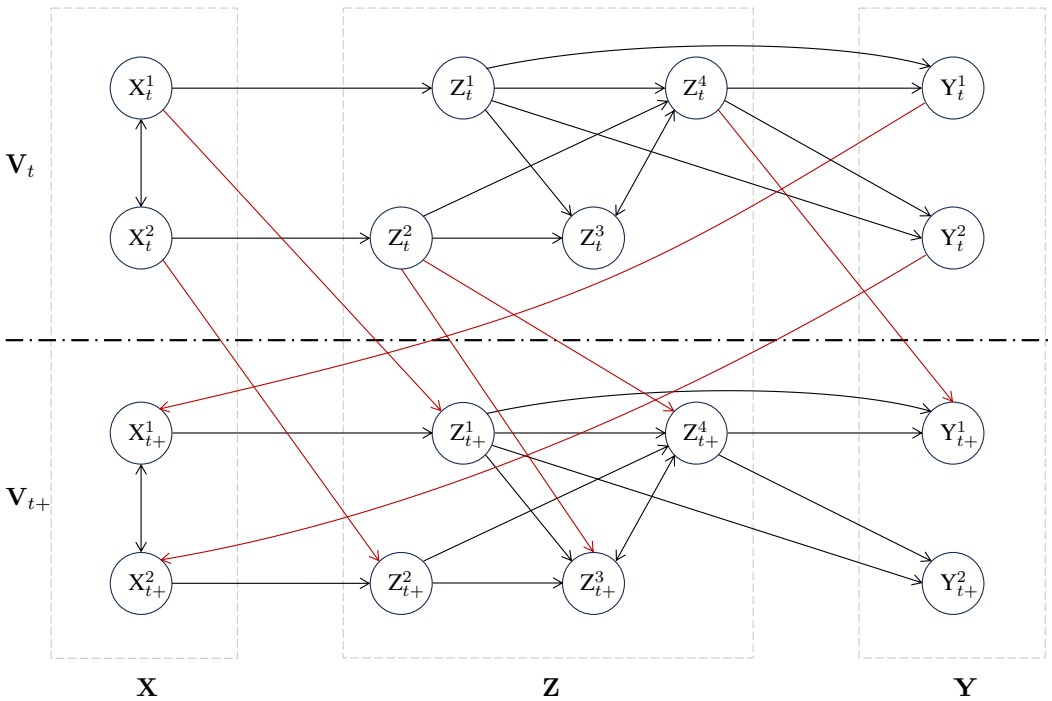

Figure 9: The rehearsal graph for synthetic data.

and the lagged influenced relations are recorded in:

$$\mathbf{B} = \begin{bmatrix} 0 & 0 & 0 & 0 & 0 & 0 & 0 & -0.6 \\ 0 & 0 & 0 & 0 & 0 & 0 & -0.6 & 0 \\ 0 & 0.6 & 0 & 0 & 0 & 0 & 0 & 0 \\ 0.6 & 0 & 0 & 0 & 0 & 0 & 0 & 0 \\ 0 & 0 & 0 & 0.7 & 0 & 0 & 0 & 0 \\ 0 & 0 & 0 & 0.2 & 0 & 0 & 0 & 0 \\ 0 & 0 & 0 & 0 & 0 & 0.3 & 0 & 0 \\ 0 & 0 & 0 & 0 & 0 & 0 & 0 & 0 \end{bmatrix}.$$

Last, the noise mean $\mathbb{E}[\varepsilon_t] \triangleq \mathbf{0}$, with $\mathrm{Cov}[\varepsilon_t] \triangleq \mathbf{\Sigma} = \begin{bmatrix} 4 & 0 & 0 & 0 & 0 & 0 & 0 & 0 \\ 0 & 4 & 0 & 0 & 0 & 0 & 0 & 0 \\ 0 & 0 & 6 & 0 & 0 & 0 & 0 & 0 \\ 0 & 0 & 0 & 6 & 0 & 0 & 0 & 0 \\ 0 & 0 & 0 & 0 & 3 & 1.6 & 0 & 0 \\ 0 & 0 & 0 & 0 & 1.6 & 6 & 0 & 0 \\ 0 & 0 & 0 & 0 & 0 & 0 & 4 & 0 \\ 0 & 0 & 0 & 0 & 0 & 0 & 0 & 12 \end{bmatrix}.$

This to say, in the demonstrated experimental results in the main paper, if $\varepsilon_t$ obeys a Gaussian distribution, then $\varepsilon_t \sim \mathcal{N}(\mathbf{0}, \mathbf{\Sigma})$; while if $\varepsilon_t$ obeys a Laplace distribution, then the scale parameter $b$ for each dimension can be computed by $2b^2 = \sigma^2$, and the covariance of mutually influenced variables can be additionally computed.

Fig. 11 reports additional results on the synthetic dataset, including the variance reduction rate w.r.t. the time window length $T$ (left, as in Thm. 3.5), the execution time w.r.t. $T$ (middle), and the excess risk $\|\tilde{\mathbf{z}}_t^{\hat{\Theta}} - \tilde{\mathbf{z}}_t^\star\|_2$ w.r.t. the number of observational samples $N$ (right, as in Thm. 3.6), which are similar to those on the Bermuda dataset in Sec. 5. Note that the execution time of both our methods $(\mathcal{O}(|\mathbf{z}||\mathbf{y}|^2 + (t_e - t_0)|\mathbf{v}||\mathbf{z}|)$ for the GMuR method and $\mathcal{O}((t_e - t_0)|\mathbf{z}||\mathbf{y}||\mathbf{v}|)$ for the FarMuR method) is significantly reduced compared to the previous methods $(\mathcal{O}\left((T+1)l|\mathbf{V}|^3\right)$ for Du et al. [7] with $l$ is the iteration times of the ulterlized optimization algorithm, and $\notin \mathcal{O}\left((T+1)|\mathbf{V}|^p\right), \forall p \in \mathbb{Z}_+$, for Qin et al. [6]). Meanwhile, as discussed in Qin et al. [6], the performance of QWZ23 [6] is unstable on this dataset because the outcome variable $\mathbf{Y}$ is not singleton, leading to an irregular execution time curve because its time complexity w.r.t. other factors, such as $|\mathbf{V}|$, could be varying.

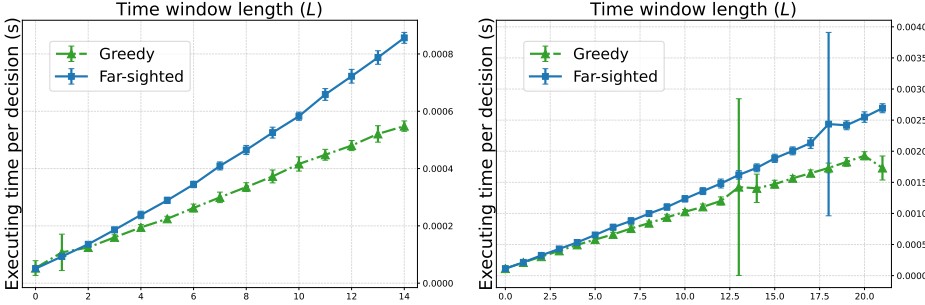

Figure 10: Results on the Bermuda dataset (left) and the synthetic dataset (right), illustrating the average executing time w.r.t. $T$ for our FarMuR method and GMuR method.

Meanwhile, Fig. 10 reports additional results on the Bermuda dataset (left) and the synthetic dataset (right), illustrating the average executing time w.r.t. $L$ for our FarMuR method and the GMuR method. This results (combining with the AUF probability in Tab. 1) illustrate the trade-off between execution time and AUF performance for the GMuR and FarMuR approaches as $L$ increases.

The variables in the Bermuda dataset are illustrated in Fig. 12, including:

- Light ($X^1$): Light levels at the bottom;

- Temp ($X^2$): Temperature at the bottom;

- Sal ($X^3$): Sea surface salinity;

- DIC ($Z^1$): Dissolved inorganic carbon of seawater;

- TA ($Z^2$): Total alkalinity of seawater;

- $\Omega_A$ ($Z^3$): Saturation with respect to aragonite in seawater;

- Nut ($Z^4$): PC1 of $NH_4$, $NiO_2 + NiO_3$, $SiO_4$;

- Chla ($Z^5$): Chlorophyll-a at sea surface;

- pHsw ($Z^6$): pH of seawater;

- $CO_2$ ($Z^7$): $P_{CO_2}$ of seawater;

- NEC ($Y^1$): Net ecosystem calcification.

The parameters associated with instantaneous influence relations (($\mathbf{A}$)) and the noise covariance matrix ($\mathrm{Cov}[\boldsymbol{\varepsilon}_t]$) are estimated by fitting least-squares linear models to the real-world data [41, 43],

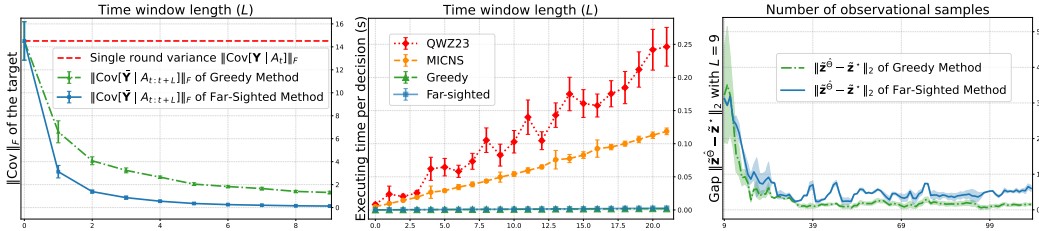

Figure 11: Results on the synthetic dataset, showing the variance reduction w.r.t. the time horizon $T$, average executing time w.r.t. $T$, and gap $\|\tilde{\mathbf{z}}_t^{\hat{\Theta}} - \tilde{\mathbf{z}}_t^{\star}\|_2$ w.r.t. the number of observational samples $N$.

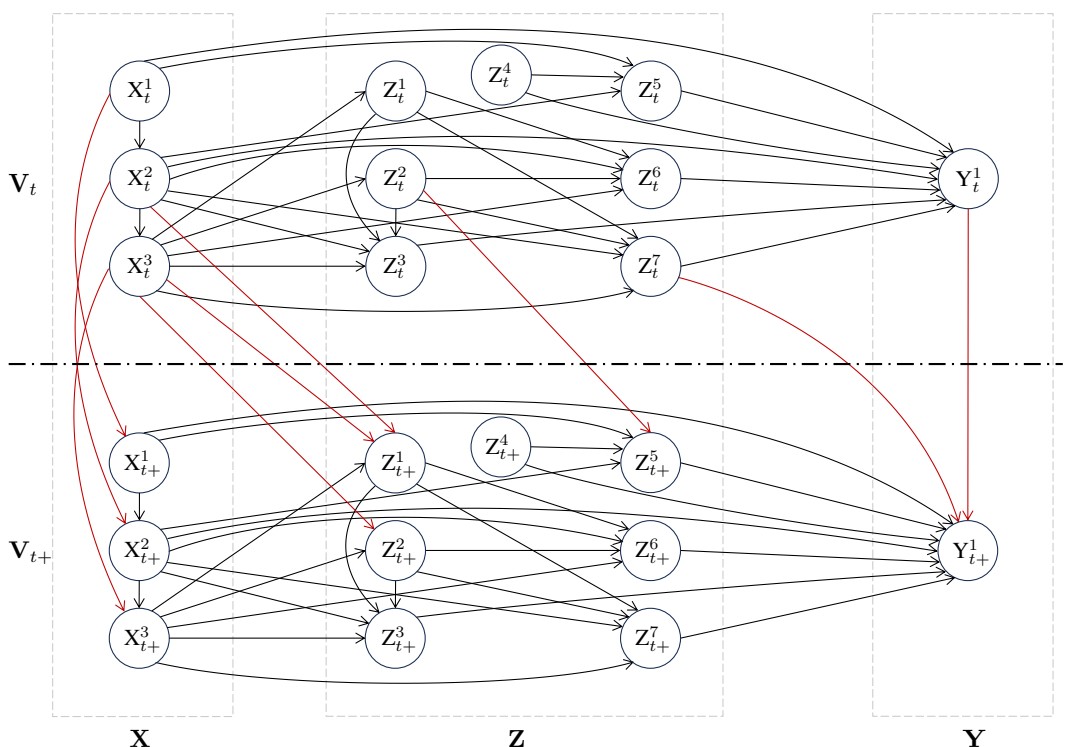

Figure 12: The rehearsal graph for Bermuda data.

while the parameters associated with the lagged influence relations ($\mathbf{B}$) are manually determined:

$$\mathbf{B} = \begin{bmatrix} 0.6 & 0 & 0 & 0 & 0 & 0 & 0 & 0 & 0 & 0 & 0 \\ 0 & 0.6 & 0 & 0 & 0 & 0 & 0 & 0 & 0 & 0 & 0 \\ 0 & 0 & 0.6 & 0 & 0 & 0 & 0 & 0 & 0 & 0 & 0 \\ 0 & -0.1 & 0.23 & 0 & 0 & 0 & 0 & 0 & 0 & 0 & 0 \\ 0 & 0 & 0.25 & 0 & 0 & 0 & 0 & 0 & 0 & 0 & 0 \\ 0 & 0 & 0 & 0 & 0 & 0 & 0 & 0 & 0 & 0 & 0 \\ 0 & 0 & 0 & 0 & 0 & 0 & 0 & 0 & 0 & 0 & 0 \\ 0 & 0 & 0 & 0 & -0.1 & 0 & 0 & 0 & 0 & 0 & 0 \\ 0 & 0 & 0 & 0 & 0 & 0 & 0 & 0 & 0 & 0 & 0 \\ 0 & 0 & 0 & 0 & 0 & 0 & 0 & 0 & 0 & 0 & 0 \\ 0 & 0 & 0 & 0 & 0 & 0 & 0 & 0 & 0 & -1.1 & 0.6 \end{bmatrix}.$$

