# OpenReview forum: "Variance-Reduced Long-Term Rehearsal Learning with Quadratic Programming Reformulation"
_NeurIPS.cc/2025/Conference — NeurIPS 2025 poster_

### Official Review · Reviewer_qTb1 · 2025-07-03

**Clarity:** 2
**Significance:** 2
**Originality:** 2
**Rating:** 4
**Confidence:** 3

**Summary:**

This paper introduces a novel approach to the Avoiding Undesired Future (AUF) problem by extending the rehearsal learning framework to long-term decision-making scenarios. It reformulates the intractable probabilistic AUF objective, maximizing the likelihood that the average of future outcomes falls within a desired region, into a tractable quadratic programming (QP) problem under mild assumptions such as linearity and log-concave noise. The authors propose two algorithms: a greedy method that optimizes each decision step independently, and a far-sighted method that accounts for future consequences to achieve better variance reduction and overall performance. Theoretical guarantees are provided for the optimality of the QP solution, variance reduction, and robustness when using estimated parameters. Empirical evaluations on synthetic and real-world datasets confirm the proposed methods significantly outperform existing approaches in both accuracy and efficiency.

**Questions:**

1. What is the practical computational limit (in terms of time horizon $𝑇$, dimensionality $|Z|$, and data size) for applying the far-sighted method in real-time or near real-time applications?
2. Can the method accommodate settings where the decision horizon $T$ is not known in advance or varies over time?
3. The paper proposes algorithms that compute optimal alterations in closed form (via QP), but it remains unclear how interpretable these solutions are. Given the practical relevance (e.g., in policy or finance), can the authors discuss whether the alteration vectors can be explained to domain experts, or whether some sparsity constraints or diagnostics could be incorporated?
4. While the paper benchmarks against previous rehearsal-based methods (QWZ23, MICNS), the experimental comparison omits broader classes of structured decision-making or robust control algorithms. Can the authors clarify why alternatives were not considered? Could a simplified comparison or qualitative discussion be added to contextualize where rehearsal-based approaches stand in the broader decision-making landscape?

**Ethical Concerns:**

["NO or VERY MINOR ethics concerns only"]

**Final Justification:**

I have reviewed their responses to other reviewers regarding non-linear settings, which have addressed some of the concerns I initially raised. Based on these clarifications and the improvements made, I am increasing my score to 4.

**Limitations:**

The authors have adequately addressed the limitations of their work in the Appendix B (Discussion) section. They explicitly acknowledge that their proposed QP reformulation may be sub-optimal under certain conditions. This reflects a clear effort to communicate the boundaries within which their methods can be expected to perform reliably.

One suggestion for improvement: It would be helpful to explicitly discuss potential negative consequences of automation in sensitive domains (e.g., economic systems, healthcare, or policy decision-making) where biased or poorly specified “undesired regions” (AUF targets) might lead to unintended harms if not designed responsibly.

**Paper Formatting Concerns:**

In line 23 and 45, The references include author name 'Zhou', but this is not the case for other references. Please use one reference type consistently.

**Quality:**

3

**Strengths And Weaknesses:**

Strengths: (1) The paper is well-grounded in theory. The authors provide a principled extension of the AUF framework to long-term settings, supported by a rigorous QP reformulation. (2) The paper is generally well-organized and clearly written. Key concepts are introduced with sufficient motivation, and the technical derivations are presented with clarity. (3) The long-term AUF setting is practically important and aligns with real-world needs, where decisions often have delayed and cumulative consequences.

Weaknesses: (1) While the QP reformulation is elegant, its optimality depends on a set of assumptions (e.g., linearity, unique targets, symmetric desired region). These are practical in many settings, but the paper could benefit from more detailed discussion and empirical evaluation of how violations of these assumptions affect performance. While a counterexample is provided, its role is limited to theoretical discussion. (2) The far-sighted method achieves superior performance but with increased computational cost. While this is acknowledged, a deeper analysis of scalability with respect to horizon length, dimensionality, and system size would improve practical understanding. (3) The comparison excludes reinforcement learning methods based on an assumption of interaction constraints. While justified, the paper would be stronger with at least some controlled comparisons or sensitivity analyses to confirm this assumption empirically, especially since many AUF problems in practice involve some feedback or batch interaction.

---

> ### Author Rebuttal · Authors · 2025-07-31
>
> We sincerely thank you for your detailed feedback and for recognizing the strengths of our work, including its strong theoretical grounding, clear presentation, and the practical importance of the long-term AUF setting. We hope our responses below fully address your concerns.
>
> ---
>
> **W1.** Empirical evaluation of the assumptions.
>
> **A1.** Thanks for this constructive suggestion. To empirically assess the impact of violating our assumptions, we conducted new experiments where the (i) *symmetric desired region* and/or (ii) *unique target* assumptions are violated. The table below presents results on a synthetic dataset (with dimensions of $X$, $Z$, and $Y$ set to 3, 2, and 2, respectively, and a decision horizon of 9). To ensure a fair comparison, when violating assumption (i), we chose a desired region with a total area similar to that in the no-violation setting.
>
> | Violation | QWZ23 [1] | MICNS [2] | Ours (Far-Sighted) |
> | - | - | - | - |
> | No | $0.342 \pm 0.017$ | $0.332 \pm 0.025$ | $0.765 \pm 0.128$ |
> | (i) | $0.225 \pm 0.011$ | $0.250 \pm 0.037$ | $0.603 \pm 0.015$ |
> | (ii) | $0.094 \pm 0.027$ | $0.052 \pm 0.020$ | $0.278 \pm 0.166$ |
> | (i)&(ii) | $0.032 \pm 0.028$ | $0.062 \pm 0.028$ | $0.192 \pm 0.123$ |
>
> These results show that although performance degrades when assumptions are violated, our method remains substantially more robust than existing approaches [1, 2], particularly when the *unique target* assumption fails. This offers strong empirical support for the practical utility of our approach, even beyond the scope of our theoretical guarantees.
>
> ---
>
> **W2&Q1.** A deeper analysis of scalability.
>
> **A2.** Thanks for raising this important point. We agree that understanding the computational scalability of our method is crucial, especially for large dimensions and long horizons. As shown in Fig. 4 (middle) of our paper, our approach is already significantly faster than prior rehearsal-based methods [1, 2] for long horizons with fixed dimensionality.
>
> To further address scalability, we propose two practical strategies:
> (i) In high-dimensional settings, heuristic methods can be developed to select a key subset of variables $Z_s \subset Z$ and approximately solve Eq. (5) over $Z_s$;
> (ii) For long horizons, the matrices ($M$, $N$, $H$ in Eq. (5)) that depend on the horizon length can be precomputed and cached offline to accelerate real-time inference.
>
> We will incorporate this discussion into the revised version. Thank you!
>
> ---
>
> **W3&Q4.** Empirical comparison with other methods.
>
> **A3.** Thanks for this insightful question. We agree that positioning our method within the broader decision-making landscape is important. While we primarily compare with rehearsal-based methods due to (i) the specific problem setting (detailed in Appx. A), and (ii) prior empirical evidence showing that alternative methods (e.g., RL) achieve limited effectiveness in this setting [1, 2], we conducted additional experiments in response to your suggestion.
>
> We include comparisons with representative RL methods (DDPG [3], SAC [4]) and a recent causal bandit algorithm [5]. For fairness, we allowed the RL agents to interact with the environment for 100 episodes—the same sample budget available to our method from observational data.
>
> | Ours | [3] | [4] | [5] |
> | - | - | - | - |
> | $0.827 \pm 0.044$ | $0.207 \pm 0.055$ | $0.182 \pm 0.039$ | $0.166 \pm 0.022$ |
>
> All experiments were conducted on the same dataset, with the same desired region $S$ and a fixed horizon 1, over 100 rounds and 10 random seeds. The table reports the AUF probability mean and variance. Our method clearly outperforms these baselines, as the rehearsal-based approach is more effective at leveraging fine-grained structural dependencies among variables and observed context $X$.
> We also observed that SAC can eventually reach comparable performance after ~6,000 training steps. However, such extensive interaction is often not feasible in our problem setting.
>
> ---
>
> **Q2.** Applicability to varying decision horizons.
>
> **A4.** Thanks for the thoughtful question.
>
> We would like to respectfully clarify that the decision horizon $T$ is a user-specified parameter (thus should not be unknown). For instance, if a trader aims to maximize profit over the next 10 months, they would simply set $T=10$.
>
> For a *varying* decision horizon setting, our main theoretical result (Thm. 3.4) can be straightforwardly applied to such scenarios. Thm. 3.4 provides a strong guarantee: for any finite future horizon $T$ and any decision step $t$, the solution to Eq.(5) is optimal in maximizing the AUF probability of the aggregated target $\sum_i Y_i/T$, assuming Ass. 3.3 holds. If decision-makers choose to specify different horizons $T$ at different steps $t$, our method remains applicable and optimal in this context. One only needs to recompute the corresponding matrices ($M$, $N$, $H$) in Eq.(5) for the specified $T$.
>
> ---
>
> **Q3.** Interpretability and sparsity constraints.
>
> **A5.** Thank you for highlighting this important issue. When the horizon is 1, the solution is easy to interpret: it directly maximizes the probability that the immediate outcome $Y$ lies within the desired region $S$. For longer horizons, interpretation becomes more subtle: the optimal alteration at each step maximizes the probability that the *average outcome* across the horizon falls in $S$.
> Using a Texas Hold'em analogy: if the goal is to win the current hand, the strategy should reflect immediate hand strength. But if the goal is to maximize winnings over 100 hands, one may initially build a specific table image (e.g., tight or loose) that improves long-term return, even if it sacrifices short-term gains.
>
> Regarding sparsity, since Eq. (5) is a convex QP, we can include sparsity-inducing regularizers (e.g., L1 penalty, as in LASSO) without losing convexity. This helps generate more interpretable alteration vectors for domain experts.
>
> We will add this discussion to the revised version. Thanks!
>
> ---
>
> **Other questions.** Suggestions and formatting.
>
> **A6.** Thanks for your detail and helpful suggestions. We will include a discussion of potential negative consequences in the Appendix and Checklist of the revised version.
>
> Regarding reference formatting: we would like to respectfully clarify that our use of `\citet` and `\citep` is intentional, following standard citation practices to distinguish between in-text (e.g., Author [X]) and parenthetical ([X]) citations. Meanwhile, we have used the `\bibliographystyle{unsrtnat}` style consistently throughout the paper. Thanks!
>
> ---
>
> We hope that our responses and additional experiments address your concerns. We would greatly appreciate your reconsideration of the paper in light of these clarifications. Thank you again for your valuable and constructive feedback.
>
> ---
>
> **Reference**
>
> [1] Rehearsal Learning for Avoiding Undesired Future. NeurIPS 2023.
>
> [2] Avoiding Undesired Future with Minimal Cost in Non-Stationary Environments. NeurIPS 2024.
>
> [3] Continuous control with deep reinforcement learning. ICLR, 2016.
>
> [4] Soft actorcritic: Off-policy maximum entropy deep reinforcement learning with a stochastic actor. ICML, 2018.
>
> [5] Additive Causal Bandits with Unknown Graph. ICML, 2023.

---

> > ### Author Response · Authors · 2025-08-06
> >
> > Dear Reviewer qTb1,
> >
> > We sincerely thank you for your time and feedback on reviewing our paper. As the discussion period is nearing its end, we would like to politely remind you and would greatly appreciate it if you could take a moment to consider our responses.
> >
> > If there are any remaining concerns about our work, we would be happy to engage in further discussion.
> >
> > Yours sincerely,
> > The Authors

---

> > ### Comment · Reviewer_qTb1 · 2025-08-09
> >
> > The authors addressed some of key weaknesses identified in the initial review. For Weakness (1), they added new experiments examining the effect of violating the symmetric desired region and unique target assumptions, as well as their combination. This is exactly the type of empirical evidence I requested.
> > For Weakness (3), they extended the comparison to include RL methods (DDPG, SAC) and a causal bandit baseline under matched interaction budgets. This situates their method more clearly within the broader decision-making landscape; however, the authors should acknowledge the limitations of RL in low-interaction settings while noting RL’s competitiveness with more interactions.
> > The paper still does not empirically test the effect of violating the linearity assumption, nor does it address scalability analysis.
> >
> > Overall, the rebuttal strengthens the empirical case for the method’s robustness and relevance, but I am maintaining my original score to reflect the remaining limitations in breadth and practical analysis.

---

> > > ### Author Response · Authors · 2025-08-09
> > >
> > > Dear Reviewer qTb1,
> > >
> > > We are pleased that our rebuttal was able to address your key concerns. Regarding the new points you raised, we would like to clarify the following:
> > > - On the **effectiveness of RL in high-interaction settings**, as noted in **A3** of our rebuttal, *We also observed that SAC can eventually reach comparable performance after ~6,000 training steps. However, such extensive interaction is often not feasible in our problem setting.*
> > > - On the **applicability of our method in non-linear settings**, we have discussed this in detail in **A1** of our response to Reviewer **JXfA** and **A1** of our response to Reviewer **2xvd**. We would politely encourage you to review those responses for a fuller explanation. Due to time constraints, we were unfortunately unable to conduct additional experiments to demonstrate this further, and we hope for your understanding.
> > >
> > > We sincerely thank you again for your time and feedback.
> > >
> > > Sincerely yours,
> > > The Authors

---

### Official Review · Reviewer_2xvd · 2025-07-03

**Clarity:** 3
**Significance:** 3
**Originality:** 2
**Rating:** 4
**Confidence:** 3

**Summary:**

The paper introduces the AUF (avoiding undesirable futures) problem for maximizing the probability of a sequence of future outcomes being in a certain set S. They work on a simple case of the problem - a linear markovian transition structure that is essentially a linear dynamical system. This allows them to formulate the problem as a QP, which they solve with both a greedy and a far-sighted method.

**Questions:**

1. Could you address the implicit questions within my discussion of the paper’s weaknesses?
2. The noise in the trajectory is non-isotropic, so I find it a bit odd that the way to maximize the probability is to optimize an objective with L2 loss, instead of an L2 loss transformed by a matrix M.

**Ethical Concerns:**

["NO or VERY MINOR ethics concerns only"]

**Final Justification:**

The empirical justification of their linearity assumption and the discussion of their relation to MPC have largely satisfied me, but I don’t think they have faithfully addressed their relation to causal RL, as I discuss in my response. So my score had gone from a 3 to a 4 but not to a 5.

**Limitations:**

The paper does not address the strength of the linearity assumption, and I think it also needs to spend some time verifying to what extent the assumption holds in its dataset (to what extent can the best fit linear transition model explain the observations in the dataset).

**Quality:**

2

**Strengths And Weaknesses:**

Strengths:
1. The paper extends rehearsal learning from single-round AUF to multi-round, long-term settings, which better reflects many practical problems (e.g., portfolio management, sequential decision-making).
2. It establishes optimality of the QP solution under reasonable (but strong) assumptions and derives variance reduction rates (Theorem 3.5) that are directly relevant for sequential AUF objectives.
3. The quadratic programming reformulation provides a computationally efficient way to handle the long-term AUF problem, avoiding sampling-based or MILP approaches that scale poorly with the dimensionality of variables.
4. The experiments, while limited, clearly show that the proposed far-sighted method achieves substantially higher AUF probabilities than both baselines and existing rehearsal learning methods across multiple datasets.

Weaknesses:
1. I think the linear assumption that crucially allows the paper to design a QP formulation of the AIUF problem is quite strong, and given the lack of experiments, it is unclear how well it holds up.
2. The linear assumption also turns this problem into a linear dynamical system, unless I am mistaken. There is extensive literature in control theory, to my understanding, which addresses the problem of controlling a linear dynamical system to let it end up in a desirable set. I am not sure if the exact problem in this work is addressed in control theory literature, but the work is worth diving into and reference.
3. Many arguments about the ineffectiveness of RL are about its stated “inability to use existing knowledge about systems.” I think this is incorrect for model-based RL. In fact, an experimental comparison with model-based RL methods under the same linear model is missing. The argument for reward drift seems incorrect to me - you can just have the reward be the indicator function of the desired set S, making it a deterministic reward with no distribution shift. The actual value of the reward now just depends on the observed Y_t.

---

> ### Author Rebuttal · Authors · 2025-07-31
>
> Thanks for your detailed feedback and for recognizing the strengths of our work! We hope our responses below can address your concerns.
>
> ---
>
> **W1&Limitation.** Concerns about the linear assumption.
>
> **A1.** Thanks for your constructive suggestion. First, we would like to clarify that our linear assumption generalizes the noise distribution to the log-concave family, which is a broader class than that considered in prior works [1, 2]. While this assumption is essential to enabling our QP reformulation, we also further discuss its generality and empirical validity as follows:
>
> - **Potential extensions to non-linear settings.**
>   The core ideas behind our algorithm are not limited to linear models. Specifically, our decision-making objective only requires that the relationship between the aggregated outcome $\sum_i Y_i/T$ and the sequence of alterations $\\{z_i\\}$ can be parameterized with additive aggregated noise. That is, if there exists a functional family $f$ such that $$\sum_i Y_i/T = f(\theta; \\{z_i\\}) + \tilde{\varepsilon},$$ for some parameters $\theta$ and aggregated noise $\tilde{\varepsilon}$, then the optimization at each decision round $k$ can proceed by minimizing the surrogate loss $$\\|f(\theta; \\{z_i\\}) - s_k\\|_2^2,$$ where $s_k$ denotes the adjusted center of the desired target region. This observation suggests that our formulation is not inherently tied to linear dynamics and may naturally extend to some general non-linear settings.
>
> - **Empirical validation of the linearity assumption.**
>   To assess the adequacy of the linear assumption, we fitted the model using least squares and evaluated its fit using standard metrics: (i) per-coordinate $R^2$ scores, (ii) normalized mean squared error (NMSE), and (iii) normalized mean absolute error (NMAE).
>
>   The linear model achieves average $R^2$ scores of `0.869/0.686`, NMSE of `0.212/0.401`, and NMAE of `0.305/0.468` on the synthetic / real-world (Bermuda) datasets, respectively. For other datasets, this validation step can also be performed to verify the suitability of the linear assumption.
>
> Taken together, these points support both the current validity and future extensibility of our formulation. We thank the reviewer again for raising this important issue.
>
> ---
>
> **W2.** Linear dynamical system and control theory.
>
> **A2.** Thanks for your insightful question. Our approach differs from classical control methods, such as online linear quadratic control (LQR) [3, 4], in several key ways:
>
> - **Different Motivations.** Traditional LQR assumes a linear dynamical system of the form $x_{t+1} = Ax_t+Bu_t+w_t$ and aims to control the state trajectory $\{x_t\}$ over time. In contrast, our goal is to maximize the probability that the **aggregated outcome** $\sum_iY_i/T$ lies within a desired region, without being concerned about the specific trajectory of $X$, $Y$, or $Z$ at individual time points.
>
> - **Distinct Formulations.** Our system cannot be reduced to the standard LQR formulation, as the alteration can also affect variables *within* the same time point. Moreover, our problem is fundamentally defined by Eq. (3), and Eq. (5) is a heuristic reformulation under certain assumptions. Even so, the resulting QP in Eq. (5) differs from that in LQR: while online LQR decomposes into per-step QPs (w.r.t. control variables like $\sum_t(x_t^\top Qx_t+u_t^\top Ru_t)$) due to additive structure, our objective involves aggregation of altered variables $z_t$ through matrix $F$, making the problem cannot be decomposed thus more challenging.
>
> - **Connection to MPC.** While our method shares a high-level similarity with Model Predictive Control (MPC) [5], in that we optimize a future sequence of actions but only execute the first, our predictive model is different from classic ones. We rely on structured dependencies (see **A3**) that go beyond the scope of standard MPC formulations.
>
> We will incorporate this discussion into the revised version. Thanks again!
>
> ---
>
> **W3.** Comparison with RL methods.
>
> **A3.** Thanks for your thoughtful comment. It appears there may be a misunderstanding of our problem setting, and we appreciate the opportunity to clarify the differences between our structural approach and conventional RL, including model-based RL.
>
> Your question highlights a key advantage of the rehearsal learning framework. Simply formulating a reward function with an indicator $\mathbb{I}(Y \in S)$ is insufficient for addressing decision problems with structural dependencies (like causal relations), especially those involving confounding. Even when the indicator function is deterministic, it is crucial to distinguish between *samples collected from different data distributions* (observation or interaction/intervention), as this distinction fundamentally impacts the decision-making process.
>
> - **Observation vs. Intervention.**
>   Our setting relies solely on *observational data* to make immediate decisions, where structural (e.g., causal) relations exist. In contrast, standard RL methods assume *interventional data* from agent-environment interactions. Treating observational samples $\langle x, z, \mathbb{I}(Y \in S) \rangle$ as if they were standard RL tuples $\langle s, a, r \rangle$ implicitly assumes that the observational and interventional distributions are identical, an invalid assumption in the presence of confounding.
>
>     *E.g.*, suppose the structure includes: $X\rightarrow Y$, $X\rightarrow Z_a\rightarrow Y$, $Z_a^\prime\rightarrow Z_u$ and $Z_a\leftarrow Z_u\rightarrow Y$, where $X$ can be viewed as the state in RL, $Z=(Z_a,Z_a^\prime)$ are two actionable variables, and $Z_u$ is an observed but unactionable confounder. Observational data might show a strong correlation between $Z = z$ and $Y\in S$ given $X=x$, leading an RL agent to learn $z$ as the optimal action incorrectly. In contrast, our method can identify the optimal action $z^\star\ne z$ without requiring costly exploration by leveraging structural information and applying *backdoor criteria* [10].
>
> - **Structural vs. Statistical Knowledge.**
>   While model-based RL can incorporate prior knowledge, it typically captures **statistical transition dynamics** (e.g., $p(s_{t+1} \mid s_t, a_t)$) rather than **fine-grained structural dependencies**. Our method leverages the latter, which is essential to distinguish causation/influence relations from purely correlation.
>
> - **New Experiments.** To empirically illustrate this distinction, we compared our method with both model-free (DDPG [6], SAC [7]) and model-based (MBPO [8]) RL agents. Each was allowed 100 environment interactions, the same sample budget available to our method from observational data.
>
>   |Method|Ours|[6]|[7]|[8]|
>   |-|-|-|-|-|
>   |AUF Probability|$0.827\pm 0.044$|$0.207\pm 0.055$|$0.182\pm 0.039$|$0.228\pm 0.027$|
>
> All experiments were run on the same dataset, with the same region $S$ and horizon 1, across 100 rounds and 10 random seeds. RL methods underperform due to their inability to exploit *structural information*, resulting in slow convergence. Notably, SAC can eventually reach comparable performance, but only after ~6,000 interactions, underscoring the sample inefficiency of RL in our context.
>
> In summary, our work is motivated by the need to incorporate structural knowledge, which is essential for making decisions in the presence of confounding. We hope this clarifies the distinction and illustrates why our approach is better suited to this class of problems than standard RL pipelines.
>
> ---
>
> **Q2.** Non-isotropic noise.
>
> **A4.** Thank you for this insightful question. Your observation is very sharp and touches upon a key aspect of our theoretical contribution.
>
> Indeed, when the target variable $Y$ is multi-dimensional, the noise in the aggregated outcome $\sum_i Y_i/T$ is possible to be non-isotropic. However, this does not affect our decision-making strategy. Let us revisit the core objective of our AUF problem: given an observation $X=x$, we aim to choose an alteration sequence $\\{z_i\\}$ to maximize the probability of $\sum_i Y_i/T \in S$. In this context, our Prop. 3.1 guarantees that the entire stochasticity of the aggregated outcome is captured by the aggregated zero-mean noise term, $F\tilde{e}$, which is **unaffected** by the choice of the choosed action variables $\\{z_i\\}$.
>
> Consequently, our optimization in Eq.(5) is performed over the sequence of action variables $\\{z_i\\}$. The choice of norm for this optimization does not alter the stochastic term $F\tilde{e}$ that governs the randomness of the outcome $\sum_i Y_i/T$. The reason our method successfully maximizes the probability of the outcome falling into the desired region $S$ is guaranteed by established results in probability theory, involving key lemmas such as Anderson's Theorem [9]. Should you be interested, a detailed proof is provided in Appx. D.3.
>
> ---
>
> We hope our detailed responses have addressed your concerns. If any points remain unclear, we would be happy to elaborate further. In light of these clarifications, we would be grateful if you would reconsider your evaluation. Thank you again for your thoughtful feedback!
>
> ---
>
> **Reference**
>
> [1] Rehearsal Learning for Avoiding Undesired Future. NeurIPS 2023.
>
> [2] Avoiding Undesired Future with Minimal Cost in Non-Stationary Environments. NeurIPS 2024.
>
> [3] Online Linear Quadratic Control. ICML 2018.
>
> [4] Naive Exploration is Optimal for Online LQR. ICML 2020.
>
> [5] Model predictive control: Theory and practice—A survey. Automatica 1989.
>
> [6] Continuous control with deep reinforcement learning. ICLR 2016.
>
> [7] Soft actor-critic: Off-policy maximum entropy deep reinforcement learning with a stochastic actor. ICML 2018.
>
> [8] When to Trust Your Model: Model-Based Policy Optimization. NeurIPS 2019.
>
> [9] The integral of a symmetric unimodal function over a symmetric convex set and some probability inequalities. American Mathematical Society 1955.
>
> [10] Causality: Models, Reasoning and Inference.

---

> > ### Author Response · Authors · 2025-08-06
> >
> > Dear Reviewer 2xvd,
> >
> > We sincerely thank you for your time and feedback on reviewing our paper. As the discussion period is nearing its end, we would like to politely remind you and would greatly appreciate it if you could take a moment to consider our responses.
> >
> > If there are any remaining concerns about our work, we would be happy to engage in further discussion.
> >
> > Yours sincerely,
> > The Authors

---

### Official Review · Reviewer_eWH7 · 2025-07-03

**Clarity:** 4
**Significance:** 3
**Originality:** 4
**Rating:** 5
**Confidence:** 4

**Summary:**

This paper introduces a new approach for "Avoiding Undesired Future" (AUF) problems in machine learning, extending existing methods to handle long-term decision-making scenarios. The authors propose a variance-reduced rehearsal learning framework that uses quadratic programming (QP) reformulation to transform intractable probabilistic optimization into a tractable one when making decisions to prevent predicted undesired outcomes over extended time horizons.

**Questions:**

1) How should practitioners handle model uncertainty and potential distribution shift over time?

2) What monitoring and adaptation strategies are recommended when the system is deployed in dynamic environments?

3) How can the approach be integrated with existing decision-making systems that may have additional constraints not captured in the current formulation?

**Ethical Concerns:**

["NO or VERY MINOR ethics concerns only"]

**Limitations:**

yes

**Paper Formatting Concerns:**

The format of the paper seems in accordance with the format stated in the website.

**Quality:**

3

**Strengths And Weaknesses:**

Strengths:

1) The paper provides rigorous theoretical guarantees including optimality of the QP reformulation under mild assumptions, variance reduction analysis, and O(1/√N) excess risk bounds

2) The quadratic programming reformulation transforms an intractable probabilistic optimization into an efficiently solvable problem

3) The far-sighted method achieves substantial performance gains (e.g., 95% vs 32% AUF probability on synthetic data)

Weaknesses:

1) The optimality guarantees require three specific assumptions (linear systems, unique targets, symmetric desired regions) that may not hold in many real-world scenarios

2) Only two datasets (one synthetic, one environmental) are used for evaluation

3) While theoretically efficient, the paper doesn't adequately address how the methods scale to high-dimensional problems or very long time horizons

---

> ### Author Rebuttal · Authors · 2025-07-31
>
> Thanks for your valuable feedback and appreciation of our work! We hope that our responses can address your concerns.
>
> ---
>
> **W1.** Concerns about the assumptions.
>
> **A1.** Thanks for the insightful question. While our theoretical guarantees (Thm. 3.3) rely on certain assumptions, our method remains effective in practice even when these assumptions are partially violated, as evidenced by the new experimental results in **A2**.
>
> The latter two assumptions, *i.e.*, *unique targets* and *symmetric desired regions*, are often satisfied in real-world applications, as they are typically determined by the decision-maker. As for the linear system assumption, we would like to clarify that we have already generalized the noise distribution beyond the Gaussian assumption used in prior works [1, 2], allowing for symmetric log-concave noise.
> Moreover, potential non-linearities could be addressed using standard techniques such as kernel methods [3, 4]. Due to space constraints, we kindly refer you to our response to Reviewer JXfA (**A1**) for a more detailed discussion of the linearity assumption.
>
> We will incorporate this discussion into the revised paper. Thanks!
>
> ---
>
> **W2.** Concerns about the experiments.
>
> **A2.** Thanks for your question. To address your concern, we conducted additional experiments on a new dataset to evaluate our method's performance when the assumptions are violated.
> The table below shows the performance on this new dataset (where dimensions of $X$, $Z$, and $Y$ are 3, 2, and 2, respectively and the horizon is set to 9) when assumption (i) symmetric desired region and/or (ii) unique target assumptions are violated. To ensure a fair comparison, when violating assumption (i) (symmetric desired region), we used a desired region with a total area similar to that of the no violation setting.
>
> | Violation | QWZ23 [2] | MICNS [3] | Ours(Far-Sighted) |
> | - | - | - | - |
> |No|$0.342\pm 0.017$ |$0.332\pm 0.025$ |$0.765\pm 0.128$ |
> | (i) |  $0.225\pm 0.011$ | $0.250\pm 0.037$ | $0.603\pm 0.015$|
> | (ii) | $0.094\pm 0.027$ | $0.052\pm 0.020$ | $0.278\pm 0.166$|
> | (i)&(ii) | $0.032\pm 0.028$ | $0.062\pm 0.028$ | $0.192\pm 0.123$|
>
> As the results show, while not explicitly discussed in previous work, violating these assumptions degrades performance of the rehearsal learning approaches, especially the unique target assumption. Although optimality is not guaranteed in these settings, our method still significantly outperforms the prior works.
>
> Finally, we would like to respectfully clarify that whenever Ass. 3.3 holds, our method is theoretically guaranteed to perform well, as established by Thms 3.4, 3.5, and 3.6.
>
> ---
>
> **W3.** Concerns about the scalability.
>
> **A3.** Thanks for your thoughtful question. Indeed, in sequential decision-making problems, both dimensionality and the time horizon can be large. We would like to highlight that, as shown in Fig. 4(mid) of our paper, our method is significantly faster than existing rehearsal learning approaches [1, 2] for a long time horizon when the dimensionality is fixed. For further reducing the executing time in decision-making, we propose two feasible directions: (i) In high-dimensional settings, heuristic methods can be developed to select a key subset of variables $Z_s \subset Z$ and approximately solve Eq. (5) over $Z_s$; (ii) For long horizons, the matrices ($M$, $N$, $H$ in Eq. (5)) that depend on the horizon length can be precomputed and cached offline to accelerate real-time inference.
>
> We will incorporate this discussion into the revised paper. Thanks!
>
> ---
>
> **Q1.** Potential distribution shift over time.
>
> **A4.** Thanks for your question. The rehearsal learning process involves two sequential steps:
> - **Step 1.** Estimate/Update the underlying system parameters $\theta$ from historical observational data.
> - **Step 2.** Use $\theta$ to construct system matrices (e.g., $A$, $B$ in Eq. (2)), and solve Eq. (5) for the decision alterations.
>
> The model uncertainty and potential distribution shifts primarily affect the parameter estimation in Step 1. To handle model uncertainty (*e.g.*, a time-varying graph structure), one could segment the data based on different structures and learn a separate set of parameters for each. To address distribution shifts of the data, if the shift is gradual, some online learning methods can be employed to update the parameters [5]; and if the shift is abrupt, a restart mechanism can be implemented [6], *i.e.*, discard the old parameters upon detecting a major shift and re-estimate them from new samples.
>
> ---
>
> **Q2.** How to deal with dynamic environments.
>
> **A5.** Thanks for your question. As noted in **A4**, our framework follows a two-step sequential process. The challenge of dynamic environments, where system parameters $\theta$ change over time, primarily pertains to the parameter estimation in Step 1. This issue has been explored in prior work on rehearsal learning [2], and similar adaptive strategies could potentially be incorporated into our framework as well.
>
> ---
>
> **Q3.** How to integrate inter additional constraints.
>
> **A6.** Thanks for the insightful suggestion. Prior works in rehearsal learning [1, 2] have considered integrating additional constraints (e.g., decision cost). If such constraints define a convex feasible set for the decision variables, they can be naturally incorporated into our QP reformulation (Eq. (5)) without compromising convexity, allowing for efficient optimization.
>
> We will incorporate this discussion into the revised paper. Thanks!
>
> ---
>
> We also take this opportunity to sincerely thank you for the careful review. Your suggestions are very important for further improving the paper. Thanks again!
>
> ---
>
> **References:**
>
> [1] Rehearsal Learning for Avoiding Undesired Future. NeurIPS 2023.
>
> [2] Avoiding Undesired Future with Minimal Cost in Non-Stationary Environments. NeurIPS 2024.
>
> [3] Kernel-based conditional independence test and application in causal discovery. UAI 2011.
>
> [4] Distinguishing Cause from Effect Using Observational Data: Methods and Benchmarks. JMLR 2016.
>
> [5] Adaptive online learning in dynamic environments. NIPS 2018.
>
> [6] Learning to optimize under non-stationarity. AISTATS 2019.

---

> > ### Author Response · Authors · 2025-08-06
> >
> > Dear Reviewer eWH7,
> >
> > We sincerely thank you for your time and feedback on reviewing our paper. As the discussion period is nearing its end, we would like to politely remind you and would greatly appreciate it if you could take a moment to consider our responses.
> >
> > If there are any remaining concerns about our work, we would be happy to engage in further discussion.
> >
> > Yours sincerely,
> > The Authors

---

> > ### Comment · Reviewer_eWH7 · 2025-08-08
> >
> > Thank you for your response and clarifications! I think this is a strong paper, and will continue to recommend acceptance.

---

> > > ### Author Response · Authors · 2025-08-09
> > >
> > > Dear Reviewer eWH7,
> > >
> > > We sincerely thank you for your time and valuable feedback, and we truly appreciate your positive evaluation and recognition of our work.
> > >
> > > Sincerely yours,
> > > The Authors

---

### Official Review · Reviewer_JXfA · 2025-07-05

**Clarity:** 3
**Significance:** 2
**Originality:** 3
**Rating:** 4
**Confidence:** 2

**Summary:**

The paper proposes to solve the sequential AUF problem of linear systems. The algorithm is to extend the current rehearsal algorithm to the sequential setting and is reduced to a QP formulation to as the core optimization problem. The work proves that it has relative tight risk bound and the variance is also well bounded by the time horizon. Empirical result show a proof-of concept numerical experiments on the algorithm on multiple linear systems.

**Questions:**

1. Can the formulation and equivalent results been extended to non-linear systems?
2. Why the dynamics have the specific graph structure?
3. What is the potential application of this problem?

**Ethical Concerns:**

["NO or VERY MINOR ethics concerns only"]

**Quality:**

3

**Strengths And Weaknesses:**

Strenghts:
1. the problem is an natural extension of previuos work and the formation reduction to QP is also reasonable.
2. Though i cannot understand the proofs in detail, it looks correct to me intuitively.
3. The experiments support the results on different linear systems, include the common assumptions on the noise distribution.

Weekness:
1. the work only focuses on the linear system, which limits its applicability. If this is not easy, empirical evidence on non-linear system should also be useful.
2. the graph structure definition of the sequential problem is not straightforward, which potentially cannot cover all problem definitions.
3. no practical application being mentioned.  it is unclear why this setting and problem is useful and worth research.

---

> ### Author Rebuttal · Authors · 2025-07-31
>
> We sincerely thank you for your detailed feedback and for recognizing the strengths of our work! We hope that our responses can adequately address your concerns.
>
> ---
>
> **W1&Q1.** Extension to nonlinear systems.
>
> **A1.** Thanks for your insightful question. The core ideas of our proposed algorithm can be extended to general nonlinear settings. In principle, as long as the relationship between (i) the mean of the aggregated outcome $\sum_i Y_i/T$ and (ii) the selected alteration sequence $\\{z_i\\}$ can be captured, *e.g.*, via a parameterized functional form $f$ such that $$\sum_i Y_i/T = f(\theta; \\{z_i\\}) + \tilde{\varepsilon},$$  for some parameter $\theta$ and residual $\tilde{\varepsilon}$, then we can turn to optimize $$\arg\min_{\\{z_i\\}}\\|f(\theta; \\{z_i\\}) - s_k\\|_2^2$$ at decision round $k$, where $s_k$ denotes the adjusted center of the desired region.
>
> Moreover, our main theoretical results can be adapted to such nonlinear settings. Taking the above example, if the surrogate loss $\\|f(\theta; \\{z_i\\}) - s_k\\|_2^2$ is convex with respect to $\\{z_i\\}$, then Thm. 3.4 still applies. This is because the proof of Thm. 3.4 is based on residual analysis, with a key step showing that the aggregated error $\tilde{\varepsilon}$ remains within the log-concave distribution family (Prop. 3.2). Therefore, as long as the nonlinearity preserves this property, our main results remain valid.
>
> We will incorporate this discussion in the revised version. Thanks!
>
> ---
>
> **W2&Q2.** Concern about the graph structure.
>
> **A2.** Thanks for the thoughtful question. We are not entirely sure if we have fully understood your concern; if our response does not address your point accurately, we would be happy to further clarify during the upcoming author-reviewer discussion period.
>
> Our framework focuses on a specific class of decision problems in which structural relations among variables can be leveraged to improve decision-making, particularly when opportunities for interaction are limited. This line of work falls under the umbrella of rehearsal learning [1, 2, 3], with connections to causal bandits and causal RL [4, 5].
>
> Furthermore, even when structural information is not explicitly available in the raw data, it can often be inferred using well-established structure learning methods [6], including in sequential [7] or cyclic [8] settings. Our method can then be applied based on the learned structure.
>
> We will incorporate this clarification into the revised paper. Thanks again!
>
> ---
>
> **W3&Q3.** Potential applications.
>
> **A3.** Thanks for the question. We would like to address it in two parts:
>
> - First, our proposed decision-making method leverages structural information. Leveraging structural information to aid decision-making has broad applications in domains such as healthcare [9], finance [10], and environmental science [11]. For instance, Simpson’s paradox [12] illustrates that accurately evaluating the effect of a treatment $X$ on an outcome $Y$ requires accounting for confounding variables, highlighting the need for causal structure in certain decision-making scenario. Recent advances in causal bandits [4, 5] and rehearsal learning [2, 3] similarly emphasize the importance of structure-aware decision-making in complex environments.
>
> - Second, our work extends prior rehearsal learning frameworks by explicitly incorporating sequential information, which is crucial in many real-world scenarios. For example, a physician prescribing medication over time for a chronic disease, or a financial institution regularly updating trading strategies. In such cases, methods that do not guarantee single-step optimality may accumulate errors over time [2, 3]. In contrast, our method guarantees optimality at each step (Thm. 3.4) and controls variance across the full sequence (Thm. 3.5). Our experimental results further support the effectiveness of our method in these settings.
>
> ---
>
> Your suggestions are constructive and important for further improving the paper. Thanks again!
>
> ---
>
> **References:**
>
> [1] Rehearsal: Learning from prediction to decision. FCS 2022.
>
> [2] Rehearsal Learning for Avoiding Undesired Future. NeurIPS 2023.
>
> [3] Avoiding Undesired Future with Minimal Cost in Non-Stationary Environments. NeurIPS 2024.
>
> [4] Bandits with unobserved confounders: A causal approach. NIPS 2015.
>
> [5] Causal bandits: Learning good interventions via causal inference. NIPS 2016.
>
> [6] DAGs with NO TEARS: Continuous Optimization for Structure Learning. NIPS 2018.
>
> [7] DYNOTEARS: Structure learning from time-series data. AISTATS 2020.
>
> [8] Discovering Cyclic Causal Models by Independent Components Analysis. UAI 2008.
>
> [9] Causal inference and counterfactual prediction in machine learning for actionable healthcare. Nature Machine Intelligence 2020.
>
> [10] The causal impact of media in financial markets. Journal of Finance 2011.
>
> [11] Environmental controls on modern scleractinian coral and reef-scale calcification. Science Advances 2017.
>
> [12] Causation, Prediction, and Search. MIT Press 2000.

---

> ### Author Response · Authors · 2025-08-06
>
> Dear Reviewer JXfA,
>
> We sincerely thank you for your time and feedback on reviewing our paper. As the discussion period is nearing its end, we would like to politely remind you and would greatly appreciate it if you could take a moment to consider our responses.
>
> If there are any remaining concerns about our work, we would be happy to engage in further discussion.
>
> Yours sincerely,
> The Authors

---

### Note · Authors · 2025-08-12

We sincerely thank all reviewers for acknowledging our contributions and confirming that our rebuttal addressed some key concerns. Below are our responses to final questions.

---

To Dear Reviewer 2xvd,

We appreciate the chance to address your causal RL concerns. We realize that *causal RL* can refer to both Structural Causal Model (SCM)-based decision-making methods from causal inference literature (our focus, Ref [26–33]) and offline policy evaluation methods under confounding from RL community (as you mentioned). Due to the overlap in terminology, we unfortunately overlooked the latter.

We briefly discuss differences between our approach and the method you mentioned:
- Use of structure. Our method incorporates graphical structure (like SCM) into the decision process, allowing information like confounding, selection bias, or ancestral relations to be *directly* utilized. Causal RL methods you mentioned typically focus on handling certain types of confounding (eg., “memoryless” confounding) during offline policy evaluation, which can be seen as special information representable within SCM.
- Reward construction. In our setting with $T=1$, the reward function can be defined as whether $Y$ lies in $S$. When $T$ increases or varies (our method can accommodate this), defining such a reward becomes hard, while this can be naturally handled within our formulation.

Thanks again for raising this point! Due to space limits, we cannot provide full discussion/references here, but we will expand our related work to cover this line of research and offer a clearer comparison in the revised version.

---

To Dear Reviewer qTb1,

We appreciate the chance to provide empirical results to address your further concerns.

- On violation of linearity. We kindly encourage you to see **A1** of our response to Reviewer 2xvd, where we present experiments showing the linearity of datasets. In addition, we implemented a setting slightly violates linearity and use a linear fit to apply our method. **Result $0.824\pm0.050$ (linear) vs. $0.778\pm0.039$ show that our method remains effective**.

- On scalability. Discussion **A2**  refers to applying the method on a subset of $Z$ or pre-storing matrices for different $T$. We include experiments to show feasibility:

||AUF Prob.|Serve Time(ms)|
|-|-|-|
|Origin|$0.824\pm0.05$|$127\pm2.4$|
|Pre-storing|Same as above|$36.7\pm5.6$|
|Subset|$0.712\pm0.14$|$23.6\pm2.3$|

Thanks again for the valuable comments!

Sincerely Yours, Authors

---

### Decision · Program_Chairs · 2025-09-17

**Decision:**

Accept (poster)

**Comment:**

This work introduces a new approach for the “Avoiding Undesired Future” problems in linear systems. The authors propose a variance reduced rehearsal learning framework based on quadratic programming that can be used to make tractable the optimization problem at the heart of AUF. The paper goes on to prove a tight risk bound and empirical results show that their methods work well via numerical experiments in linear systems. The reviewers agreed this work meets the requirements for publication at Neurips.